# Multivalency, autoinhibition, and protein disorder in the regulation of interactions of dynein intermediate chain with dynactin and the nuclear distribution protein

Kayla A Jara[1], Nikolaus M Loening[2], Patrick N Reardon[1,3], Zhen Yu[1], Prajna Woonnimani[1], Coban Brooks[1], Cat H Vesely[1], Elisar J Barbar[1]*

[1]Department of Biochemistry and Biophysics, Oregon State University, Corvallis, United States; [2]Department of Chemistry, Lewis & Clark College, Portland, United States; [3]Oregon State University NMR Facility, Corvallis, United States

**Abstract** As the only major retrograde transporter along microtubules, cytoplasmic dynein plays crucial roles in the intracellular transport of organelles and other cargoes. Central to the function of this motor protein complex is dynein intermediate chain (IC), which binds the three dimeric dynein light chains at multivalent sites, and dynactin p150[Glued] and nuclear distribution protein (NudE) at overlapping sites of its intrinsically disordered N-terminal domain. The disorder in IC has hindered cryo-electron microscopy and X-ray crystallography studies of its structure and interactions. Here we use a suite of biophysical methods to reveal how multivalent binding of the three light chains regulates IC interactions with p150[Glued] and NudE. Using IC from *Chaetomium thermophilum*, a tractable species to interrogate IC interactions, we identify a significant reduction in binding affinity of IC to p150[Glued] and a loss of binding to NudE for constructs containing the entire N-terminal domain as well as for full-length constructs when compared to the tight binding observed with short IC constructs. We attribute this difference to autoinhibition caused by long-range intramolecular interactions between the N-terminal single α-helix of IC, the common site for p150[Glued], and NudE binding, and residues closer to the end of the N-terminal domain. Reconstitution of IC subcomplexes demonstrates that autoinhibition is differentially regulated by light chains binding, underscoring their importance both in assembly and organization of IC, and in selection between multiple binding partners at the same site.

**\*For correspondence:**
Elisar.Barbar@oregonstate.edu

**Competing interest:** The authors declare that no competing interests exist.

## Editor's evaluation

This is a scholarly paper. The identification of an auto-inhibitory mechanism of the dynein intermediate chain provides an important contribution to our understanding of dynein assembly. It further illustrates the plethora of regulatory mechanisms attainable by IDPs and the range of different biophysical techniques that can be used to elucidate them.

## Introduction

Dynein and kinesins are microtubule-based motor proteins that are responsible for the transport of cellular cargos such as membranes, RNAs, proteins, and viruses (*Reck-Peterson et al., 2018*; *Raaijmakers and Medema, 2014*). Whereas kinesins typically move toward the plus-end of microtubules,

**eLife digest** Motor proteins are the freight trains of the cell, transporting large molecular cargo from one location to another using an array of 'roads' known as microtubules. These hollow tubes are oriented, with one extremity (the plus-end) growing faster than the other (the minus-end). While over 40 different motor proteins travel towards the plus-end of microtubules, just one is responsible for moving cargo in the opposite direction. This protein, called dynein, performs a wide range of functions which must be carefully regulated, often through changes in the shape and interactions of various dynein segments.

The intermediate chain is one of the essential subunits that form dynein, and it acts as a binding site for a range of molecular actors. In particular, it connects the three other dynein subunits (known as the light chains) to the dynein heavy chain containing the motor domain. It also binds to two non-dynein proteins: NudE, which helps to organise microtubules, and the p150[Glued] region of dynactin, a protein required for dynein activity. Despite their distinct roles, p150[Glued] and NudE attach to the same region of the intermediate chain, a highly flexible 'unstructured' segment which is difficult to study. How the binding of p150[Glued] and NudE is regulated has therefore remained unsolved.

In response, Jara et al. decided to investigate how the three dynein light chains may help to control interactions between the intermediate chain and non-dynein proteins. They used more stable versions of dynein, NudE and dynactin (from a fungus that grows at high temperatures) to produce the various subcomplexes formed by the intermediate chain, the three dynein light chains, and parts of p150[Glued] and NudE. A suite of biophysical techniques was applied to study these structures, as they are challenging to capture using traditional approaches. This revealed that the unstructured region of the intermediate chain can fold back on itself, bringing together its two extremities; such folding blocks the p150[Glued] and NudE binding site. This obstruction is cleared when the light chains bind to the intermediate chain, demonstrating how these three subunits can regulate dynein activity.

In humans, mutations in dynein are associated with a range of serious neurological and muscular diseases. The work by Jara et al. brings new insight into the way this protein works; more importantly, it describes how to combine several biophysical techniques to study non-structured proteins, offering a blueprint that is likely to be relevant for a wide range of scientists.

dynein moves toward the minus-end. Mutations in the dynein transport machinery have been linked to neurological and neurodevelopmental diseases such as Perry syndrome, spinal muscular atrophy-lower extremity predominant, Charcot-Marie-Tooth disease, lissencephaly, and other malformations of cortical development (*Lipka et al., 2013*; *Harada et al., 1998*). Importantly, a single dynein (cytoplasmic dynein-1) performs a wide variety of functions whereas more than 40 kinesins are used to carry out a comparable number of functions (*Reck-Peterson et al., 2018*), underscoring dynein's significance and also suggesting a high level of intricate regulation processes that dictate interactions within and between dynein subunits. One vital subunit is dynein intermediate chain (IC), which plays key roles in modulating dynein interactions (*Reck-Peterson et al., 2018*; *Morgan et al., 2011*; *Jie et al., 2015*; *Makokha et al., 2002*; *Nyarko and Barbar, 2011*). Like all other dynein subunits, there are two copies of IC in the dynein motor protein complex. These two IC subunits connect the three dynein light chains (Tctex, LC8, and LC7) to the heavy chain (HC) and serve as the primary binding site of multiple non-dynein proteins essential for dynein regulation, such as the p150[Glued] subunit of dynactin and nuclear distribution protein (NudE) (*Figure 1*).

Despite its essential role in dynein function, high-resolution structural information of IC interactions is limited due to its long, disordered N-terminal domain, which hinders studies by methods such as X-ray crystallography and cryo-electron microscopy (cryo-EM). These flexible, intrinsically disordered domains often facilitate the formation of multi-protein macromolecular assemblies (*Cortese et al., 2008*; *Clark et al., 2016*; *Fung et al., 2018*) and thus, much of what remains unknown about dynein structure and regulation could be attributed to IC and other subunits containing intrinsically disordered regions (IDRs), such as dynein light IC. The lack of a folded stable structure provides highly accessible regions for post-translational modifications and ideal locations for short linear binding motifs, and thus IDRs are often promiscuous and uniquely regulated in their binding interactions (*van der Lee et al., 2014*; *Dunker et al., 2001*). Additionally, IDRs in IC interact with their partners multivalently, which

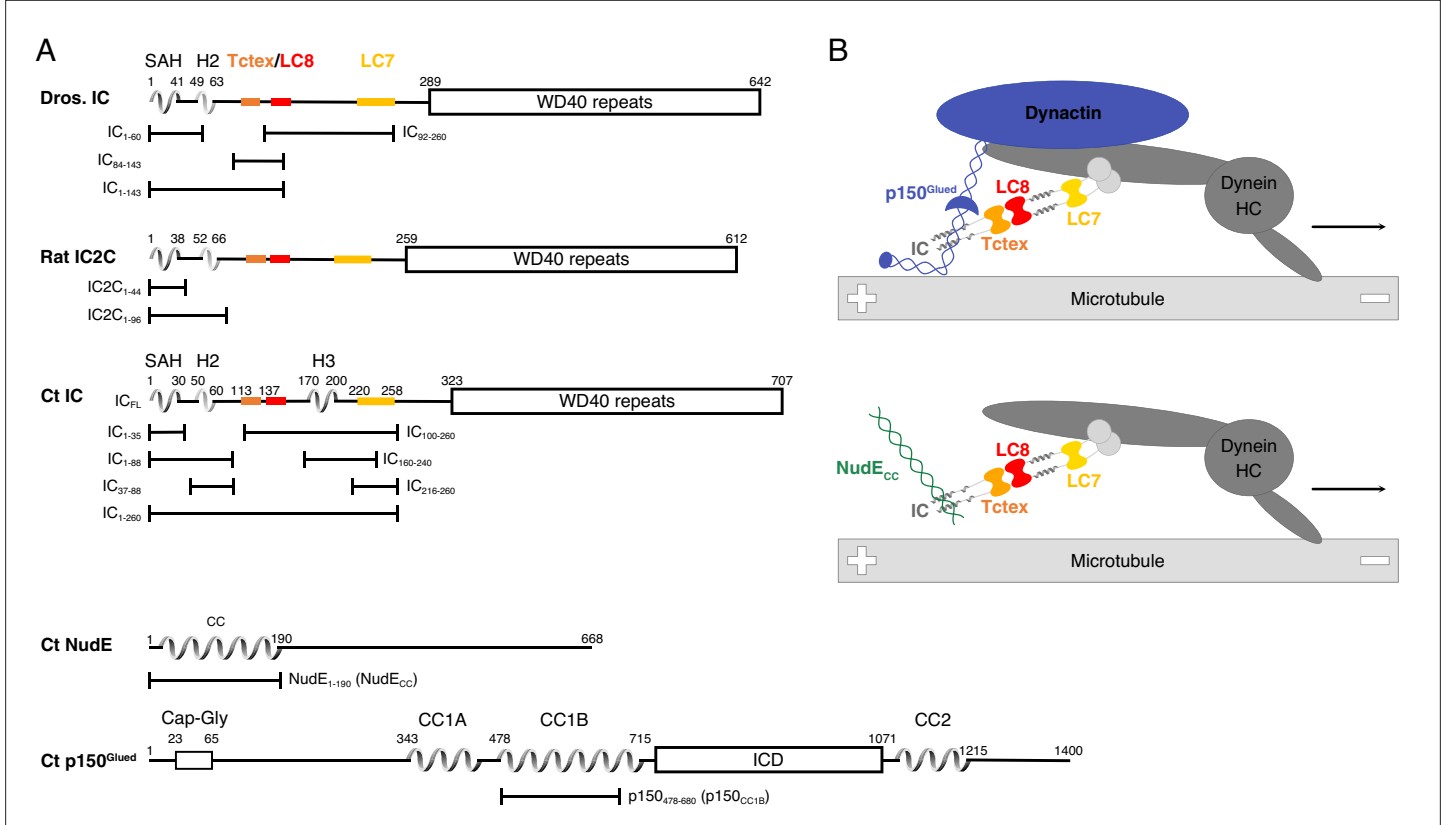

**Figure 1.** Domain architecture for dynein intermediate chain (IC), dynactin p150[Glued], and nuclear distribution protein (NudE). (**A**) Domain architecture diagrams for IC from *Drosophila melanogaster* (Dros. IC) and *Rattus norvegicus* (Rat IC2C) and constructs used in earlier work are provided for comparison. Proteins and constructs used in this work are from *Chaetomium thermophilum* (Ct). All ICs have an N-terminal single α-helix (SAH), followed by either a transient/nascent or folded second helix (H2). In Ct, there is an additional helix (H3). The Tctex (orange), LC8 (red), and LC7 (yellow) binding sites are well characterized in Dros. IC and Rat IC2C, and their position in Ct were predicted based on sequence and structure comparison. The C-terminal domain is predicted to contain seven WD40 repeats. The Ct constructs IC[FL], IC[1-88], IC[37-88], IC[1-260], IC[100-260], IC[140-260], IC[216-260], and IC[216-260] are used in this paper; the IC[1-35] construct was used in prior work. Ct p150[Glued] is predicted to have a Cap-Gly domain near the N-terminus, and two coiled-coil (CC) domains, CC1 and CC2, that are separated by an intercoil domain. CC1 is further divided into two regions called CC1A and CC1B. p150[478-680] (p150[CC1B]) is the construct used in this work. Ct NudE is predicted to have an N-terminal CC region followed by disorder. NudE[1-190] (NudE[CC]) is the construct used in this work. (**B**) Contextual models of dynein with the heavy chains (HC) crudely shown in dark gray. IC in the subcomplex (light gray) is shown in the same orientation as the domain architecture schematic in panel A. The top model depicts the interaction between the p150[Glued] subunit of dynactin (blue) and the SAH and H2 regions of IC while the bottom model depicts the interaction between NudE[CC] (green) and the SAH region of IC. In both models, dynein is a processive motor traveling toward the minus end of a microtubule, and IC is shown with the homodimeric dynein light chains: Tctex (orange), LC8 (red), and LC7 (yellow).

The online version of this article includes the following figure supplement(s) for figure 1:

**Figure supplement 1.** Intermediate chain (IC) sequence alignment.

compared to monovalent interactions involve linked associations of ligands binding to multiple sites and often confer increased binding affinity (*Barbar and Nyarko, 2015*; *Clark et al., 2015*; *Forsythe and Barbar, 2021*).

The primarily disordered N-terminal domain of IC (N-IC) (*Makokha et al., 2002*) contains a single α-helix (SAH) and a short helix (H2), each of which is followed by disordered linker regions (*Jie et al., 2015*), whereas the C-terminal domain (C-IC) consists of seven WD40 repeats that fold into a β-propeller and binds helical bundles 4 and 5 of HC (*Urnavicius et al., 2018*; *Zhang et al., 2017*; *Figure 1A*). Free N-IC is monomeric but, upon binding the homodimeric dynein light chain subunits (Tctex, LC8, and LC7), it dimerizes to form a subcomplex that is best described as a polybivalent scaffold (*Nyarko and Barbar, 2011*; *Barbar and Nyarko, 2015*; *Lo et al., 2001*; *Mok et al., 2001*; *Susalka et al., 2002*; *Hall et al., 2010*; *Figure 1B*). In the formation of this subcomplex, the first binding event pays the entropic cost for subsequent bivalent binding events (*Barbar and Nyarko, 2015*; *Nyarko and*

*Barbar, 2011*; *Hall et al., 2009*), and the enhancement of subsequent binding events is modulated by the length of the disordered linkers separating the binding sites (*Hall et al., 2009*; *Reardon et al., 2020*). Such a mechanism has been well described for the IC/Tctex/LC8 complex, for which a three-residue linker separates the Tctex and LC8 binding sites. In this complex, binding of one light chain to IC results in a 50-fold enhancement in the affinity for binding the other light chain (*Hall et al., 2009*). Each homodimeric light chain has a corresponding binding site on IC that is initially disordered but forms β-strands (for Tctex and LC8) or an α-helix (for LC7) when bound and incorporated into the fold of their respective ligand. The assembly of monomeric N-IC and the homodimeric light chains is such that, when bound, folding occurs only at the protein-protein interfaces (*Hall et al., 2010*; *Hall et al., 2009*) while the remaining linker regions stay completely disordered (*Nyarko and Barbar, 2011*; *Benison et al., 2006*; *Morgan et al., 2021*). In this work, we demonstrate the importance of the flexibility of these disordered linker regions separating the three dimeric light chains in regulating interactions of IC with non-dynein binding partners.

A common site on N-IC links dynein to the non-dynein regulatory proteins dynactin and NudE (*Morgan et al., 2011*; *Jie et al., 2015*; *Vallee et al., 2012*; *King et al., 2003*; *Jie et al., 2017*; *Nyarko et al., 2012*; *Loening et al., 2020*). Dynactin is a multisubunit complex that binds dynein with its largest subunit, p150$^{Glued}$, and this interaction is required for the recruitment of cargo, dynein processivity, and correct spindle formation in cell division (*King et al., 2003*; *Gill et al., 1991*; *Kardon and Vale, 2009*; *Schroer, 2004*; *Splinter et al., 2012*; *Carter et al., 2016*; *Urnavicius et al., 2015*). NudE, on the other hand, regulates dynein recruitment to kinetochores and membranes, centrosome migration, mitotic spindle orientation, and binds LIS1 (Lissencephaly-1 homolog) (*Kardon and Vale, 2009*; *Feng and Walsh, 2004*; *Wainman et al., 2009*; *Liang et al., 2007*; *McKenney et al., 2011*; *Wang and Zheng, 2011*; *Zyłkiewicz et al., 2011*; *Yamada et al., 2008*). It is well recognized that IC partnering with either p150$^{Glued}$ or NudE impacts the regulation of the dynein complex as this dictates its interactions with adaptors and cargo (*Reck-Peterson et al., 2018*). However, the molecular processes underlying which regulator is bound at any given time are still unclear. Previously, we showed using homologs from multiple species (rat, *Drosophila*, yeast) that the coiled-coil domains of p150$^{Glued}$ and NudE each bind IC at the same site (the SAH region) (*Morgan et al., 2011*; *Jie et al., 2017*; *Nyarko et al., 2012*) whereas for homologs from a filamentous fungus (*Chaetomium thermophilum* [Ct]) the binding of IC and p150$^{Glued}$ also involves binding of the H2 region (*Loening et al., 2020*).

Dynein is seemingly a perfect candidate for structural characterization by cryo-EM, as the motor domains are large and symmetric. Extensive cryo-EM structure analyses show that dynactin reorients the motor domains of dynein parallel to the microtubules prior to binding (*Zhang et al., 2017*; *Chowdhury et al., 2015*). Additionally, various adaptors can recruit a second dynein to dynactin for faster movement (*Urnavicius et al., 2018*; *Chaaban and Carter, 2022*). However, a recurring theme in these studies is that the flexibility of N-IC limits details of its structure and binding interactions. As a result, residue-specific studies on recombinant IC have so far been limited to short fragments and to conditions far removed from native biological systems. Using the combined data from EM structures and studies on short constructs, we present a simplified model that specifically emphasizes the interactions of IC to aid in visualizing the assembled N-IC subcomplex bound to either p150$^{Glued}$ or NudE, providing context for the work we focus on here (*Figure 1B*).

The interactions of short fragments of N-IC with dynein light chains and non-dynein proteins from multiple different species (rat, *Drosophila*, yeast) demonstrate that NudE and p150$^{Glued}$ compete for the same binding site; however, a mechanism for IC partner selection and the importance of bivalency in IC subcomplex assembly have remained elusive (*Morgan et al., 2011*; *Jie et al., 2015*; *Hall et al., 2010*; *Hall et al., 2009*; *Jie et al., 2017*; *Nyarko et al., 2012*; *Wainman et al., 2009*; *Liang et al., 2007*; *Siglin et al., 2013*). We recently introduced Ct, a thermophilic filamentous fungus, as a more tractable system for dynein structural studies (*Loening et al., 2020*). The Ct IC$_{1-260}$ (residues 1–260) construct used in this work is far longer than Ct, *Drosophila*, and rat constructs that we have previously studied (*Figure 1A*, *Figure 1—figure supplement 1*). Utilizing the entire N-terminal domain of IC allows, for the first time, characterization of IC interactions in the context of its assembly with the light chains, as has been shown vital for other disorder-regulated systems (*Clark et al., 2015*; *Bugge et al., 2020*; *Berlow et al., 2017*; *Chen et al., 2019*; *Clark et al., 2018*). However, none of the reported systems has the level of complexity (an assembly of five unique proteins) as the dynein system explored here. From our studies of both IC$_{1-260}$ as well as full-length IC, (1) we describe the

first recombinant expression and reconstitution of the polybivalent scaffold formed from IC bound by all three light chains (Tctex, LC8, and LC7), as well as by coiled-coil domains of p150$^{Glued}$ or NudE (*Figure 1B*), (2) we identify long-range tertiary contacts between residues in the LC7 binding site and residues in the N-terminal region (SAH) that inhibit binding to non-dynein proteins, (3) we show how assembly with the light chains relieves this autoinhibition and regulates binding of IC to p150$^{Glued}$ and NudE, (4) we demonstrate for the first time the essential role of light chains in both assembly and regulation of the full-length IC, and (5) we showcase how a suite of biophysical techniques are essential to effectively study intrinsically disordered proteins (IDPs) and their assemblies.

## Results

### Ct p150$_{CC1B}$ and NudE$_{CC}$ are dimeric, while Ct IC$_{1-260}$ is monomeric

Studies of interactions of IC constructs with light chains show that two monomeric IC chains are brought together by the dimeric light chains to create a 'ladder-like' polybivalent scaffold (*Barbar and Nyarko, 2015*; *Nyarko and Barbar, 2011*; *Hall et al., 2009*; *Morgan et al., 2021*). Here we use a construct of IC that includes all the binding sites for the light chains and non-dynein proteins, the full-length light chains, and coiled-coil domains of non-dynein proteins and employ multiple techniques to determine their association states. Sedimentation velocity analytical ultracentrifugation (SV-AUC), which gives information about a protein's mass and shape in solution, was used to determine each protein's heterogeneity and size. A larger sedimentation coefficient (S) indicates a protein with a larger mass and/or a protein with a smaller, shape-dependent, frictional ratio (*Cole et al., 2008*). SV-AUC showed that IC$_{1-260}$, the three light chains, and non-dynein proteins p150$_{CC1B}$ and NudE$_{CC}$ all have similar sedimentation coefficients (in the 2–3 S range, *Figure 2A*). Furthermore, all proteins show a single sharp peak in the $c(s)$ distribution, indicating that they are homogeneous in solution. In comparison to the sedimentation coefficients for IC$_{1-260}$ and the light chains, both p150$_{CC1B}$ and NudE$_{CC}$ have smaller sedimentation coefficients than would typically be expected for globular proteins with their respective dimeric masses. However, as sedimentation coefficients also depend on shape, the smaller values observed for p150$_{CC1B}$ and NudE$_{CC}$ are consistent with their predicted coiled-coil structures (which result in elongated, less-compact, rod-like shapes), causing slower sedimentation than if they were globular.

Sedimentation equilibrium analytical ultracentrifugation (SE-AUC) confirmed the dimeric coiled-coil state of p150$_{CC1B}$ and NudE$_{CC}$. SE-AUC data fit to a monomer-dimer model with dimerization dissociation constants of 0.03 μM and 0.20 μM for p150$_{CC1B}$ and NudE$_{CC}$, respectively, indicate that both are strong dimers and that, of the two, p150$_{CC1B}$ is the tighter dimer (*Figure 2B*). Circular dichroism (CD) spectra acquired at temperatures ranging from 5 to 60°C (*Figure 2C*) show that both proteins unfold with a midpoint temperature between 40 and 50°C. Also of note, the coiled-coil structures of p150$_{CC1B}$ and NudE$_{CC}$ are confirmed by their CD spectra which show helical structures and values for 222/208 ratios larger than 1 (*Figure 2C*).

To confirm our expectation that IC$_{1-260}$ is a monomer in solution, we used size exclusion chromatography (SEC) with multi-angle light scattering (MALS) detection (*Figure 2D*). The measured mass of approximately 29.5 kDa based on the MALS data is consistent with the expected mass of 29.2 kDa for an IC$_{1-260}$ monomer.

### Ct IC$_{1-260}$ is stabilized by long-range contacts

The N-IC from Ct has a domain architecture with structural elements that are similar to experimentally characterized ICs: an N-terminal SAH, a nascent helix 2 (H2), and long disordered linkers (*Figure 1A*). Ct IC is unique, however, in also including a strongly predicted third helix (H3) corresponding to residues 170–200 (*Buchan and Jones, 2019*; *Muñoz and Serrano, 1997*; *Muñoz and Serrano, 1994*; *Muñoz and Serrano, 1995a*; *Muñoz and Serrano, 1995b*; *Nucleic Acids Research, 2021*; *Figure 3A*, *Figure 3—figure supplement 1*). To validate this predicted secondary structure, and determine its impact on global stability, we acquired CD spectra of various IC constructs: IC$_{1-260}$, IC$_{1-88}$ (containing SAH and H2), IC$_{100-260}$ (containing linker, H3, and the LC7 binding site), and IC$_{160-240}$ (containing H3 and shorter linker). All the CD spectra show two minima around 208 and 222 nm, which are indicative of the presence of α-helical secondary structure (*Greenfield, 2006*; *Figure 3B–E*). The estimated fractional helicity (*Wei et al., 2014*) values for these constructs at 5°C are in the range of 20–35%,

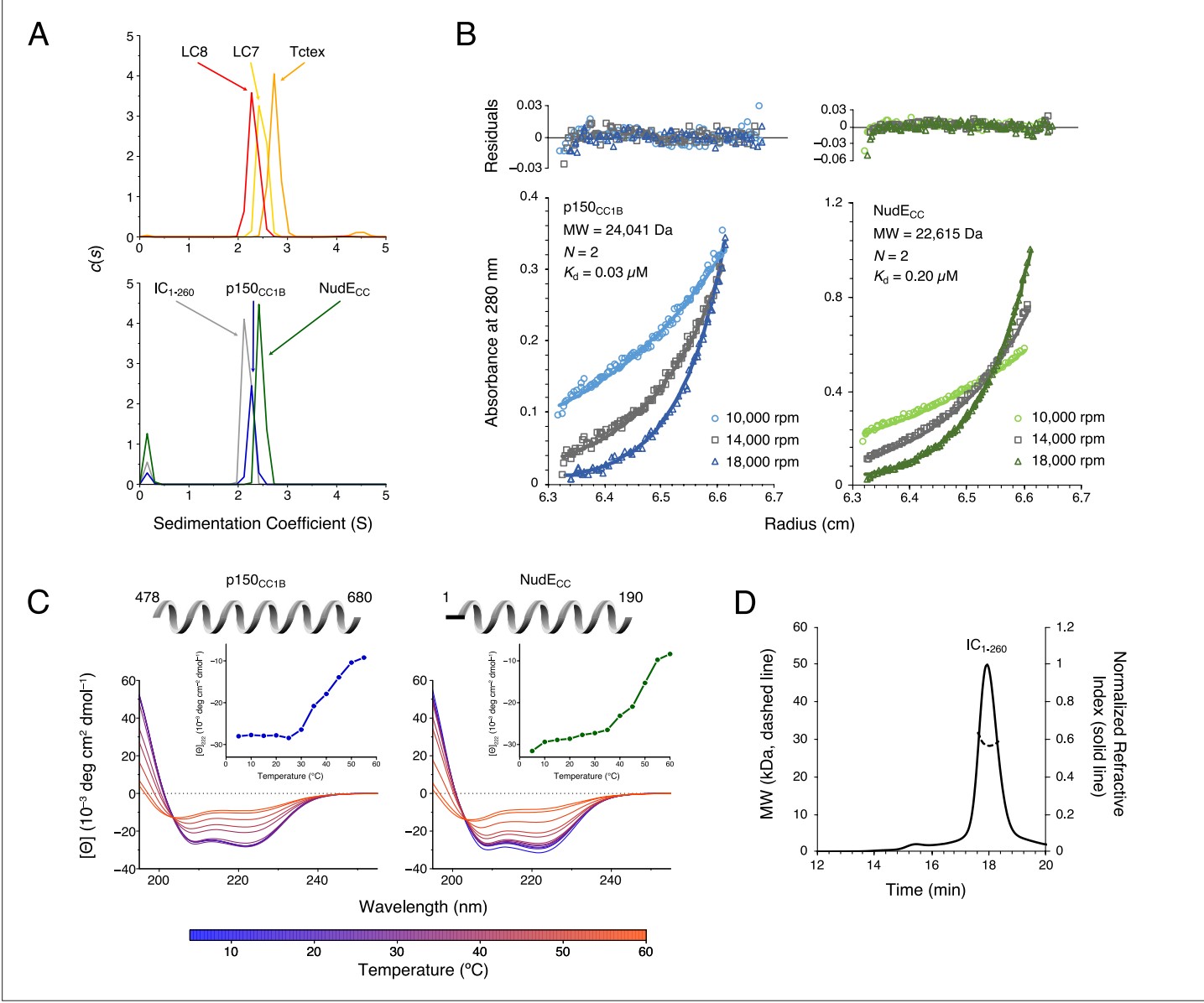

**Figure 2.** *Chaetomium thermophilum* (Ct) p150$_{CC1B}$, NudE$_{CC}$, and dynein light chains are dimeric, whereas Ct IC$_{1-260}$ is monomeric. (**A**) Sedimentation velocity analytical ultracentrifugation profiles for LC8 (red), LC7 (yellow), and Tctex (orange) (top), and, IC$_{1-260}$ (gray), p150$_{CC1B}$ (blue), and NudE$_{CC}$ (green) (bottom). All samples were at protein concentration of 30 µM. (**B**) Sedimentation equilibrium analytical ultracentrifugation data for p150$_{CC1B}$ (blue) and NudE$_{CC}$ (green) at three speeds (10,000, 14,000, and 18,000 rpm). Data were fit to a monomer-dimer binding model. The quality of the fits to this model is reflected by the plots of the residuals on top. The monomeric masses determined by fitting this data compare very well to the masses expected based on the sequences for the constructs. The stoichiometry (*N*) values of 2 indicate that both p150$_{CC1B}$ and NudE$_{CC}$ are dimers in solution. (**C**) Circular dichroism spectra of p150$_{CC1B}$ and NudE$_{CC}$ acquired at temperatures in the 5–60°C range. The shape of the spectra for both p150$_{CC1B}$ and NudE$_{CC}$ indicates α-helical secondary structure, and the 222/208 ratios (1.04 and 1.00 for p150$_{CC1B}$ and NudE$_{CC}$, respectively) are consistent with coil-coiled structures. Inset graphs show the molar ellipticity at 222 nm as a function of temperature. (**D**) Multi-angle light scattering of IC$_{1-260}$ gives an estimated mass of 29.5 kDa, which indicates that, on its own, IC$_{1-260}$ exists as a monomer in solution (calculated mass of monomer is 29.2 kDa).

The online version of this article includes the following source data for figure 2:

**Source data 1.** Source files for sedimentation velocity analytical ultracentrifugation, sedimentation equilibrium analytical ultracentrifugation, circular dichroism, and size exclusion chromatography with multi-angle light scattering data.

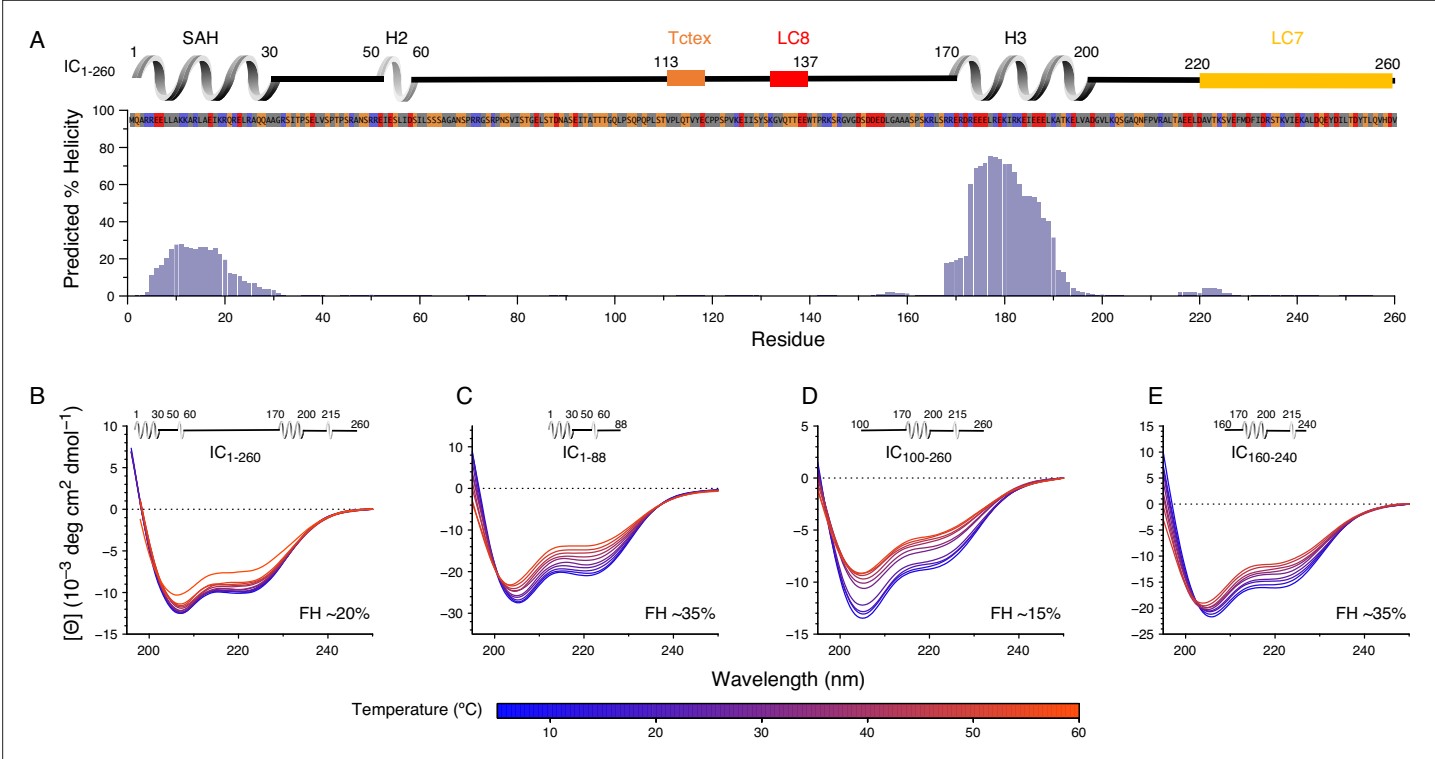

**Figure 3.** Secondary structure and thermal stability of *Chaetomium thermophilum* (Ct) intermediate chain (IC). (**A**) Agadir prediction for IC$_{1-260}$ showing the percent helicity by residue (purple). Shown above the plot is a schematic structure for IC$_{1-260}$ with labels for single α-helix (SAH), H2, and H3 above the helical structure. The sites for lights chain binding are also indicated. The amino acid sequence under the schematic is colored by amino acid type: hydrophobic (gray), positive (red), negative (blue), and neutral (orange). Variable temperature CD spectra of (**B**) IC$_{1-260}$, (**C**) IC$_{1-88}$, (**D**) IC$_{100-260}$, and (**E**) IC$_{160-240}$. The shapes of the spectra for all constructs indicate a mixture of α-helical secondary structure and regions of intrinsic disorder. Loss in structure, or lack thereof, over a temperature range of 5–50°C (blue for lowest, red for highest) indicates how each construct varies in stability and indicates that IC$_{1-260}$ is the most thermally stable. The fractional helicity (FH) of each construct at 5°C was calculated based on the experimentally observed mean residue ellipticity at 222 nm as explained in the Methods section.

The online version of this article includes the following source data and figure supplement(s) for figure 3:

**Source data 1.** Source files for IC$_{1-260}$ helicity prediction and CD data.

**Figure supplement 1.** Predicted percent helicity in intermediate chain (IC) across species.

**Figure supplement 1—source data 1.** Source file for helix prediction of intermediate chain across species.

**Figure supplement 2.** *Chaetomium thermophilum* (Ct) IC$_{1-88}$ and Ct IC$_{100-260}$ binding by sedimentation velocity analytical ultracentrifugation (SV-AUC).

**Figure supplement 2—source data 1.** Source file for sedimentation velocity analytical ultracentrifugation (SV-AUC) and size exclusion chromatography with multi-angle light scattering (SEC-MALS) of binding interactions between IC$_{1-88}$ and IC$_{100-260}$.

which matches the fraction of predicted helical residues in each construct (*Figure 3A*). Notably, the longest IC construct (IC$_{1-260}$) is the most thermostable of the four and resists unfolding until above 50°C. Comparatively, IC$_{1-88}$, IC$_{160-240}$, and IC$_{100-260}$ exhibit some loss in secondary structure significantly below 50°C. In particular, IC$_{100-260}$ appears to be the least stable of the three, with complete loss of helical structure around 40°C. From this, it is reasonable to hypothesize that some degree of tertiary contacts may exist only within IC$_{1-260}$ and underlie the increase in its structural stability and cooperative unfolding.

To further probe possible tertiary contacts within IC$_{1-260}$, SV-AUC was employed to determine if there is binding between the IC$_{1-88}$ and IC$_{100-260}$ constructs (*Figure 3—figure supplement 2A*). IC$_{1-260}$ has a sedimentation coefficient of 2.2 S, whereas IC$_{100-260}$ has a smaller sedimentation coefficient of 1.5 S due to its lower mass and its expected elongation compared to IC$_{1-260}$ (*Figure 3—figure supplement 2B*). Upon addition of IC$_{1-88}$ at a 1:2 molar ratio (IC$_{100-260}$:IC$_{1-88}$), a complex with a sedimentation coefficient of 2 S is formed, indicating a strong interaction between the two constructs; a peak for the excess of IC$_{88}$ is not observed because IC$_{1-88}$ does not absorb at 280 nm. The IC$_{1-88}$/IC$_{100-260}$

complex has a slightly smaller sedimentation coefficient than that of IC$_{1-260}$, which can be explained by a greater degree of elongation for this complex compared to IC$_{1-260}$. SEC-MALS determined a mass of 30.3 kDa for the IC$_{1-88}$/IC$_{100-260}$ complex which matches the expected mass of 30.6 kDa for a 1:1 complex (*Figure 3—figure supplement 2C*). Together, these data confirm the presence of strong tertiary contacts within IC$_{1-260}$ and explain the increase in its stability compared to smaller IC constructs (*Figure 3*).

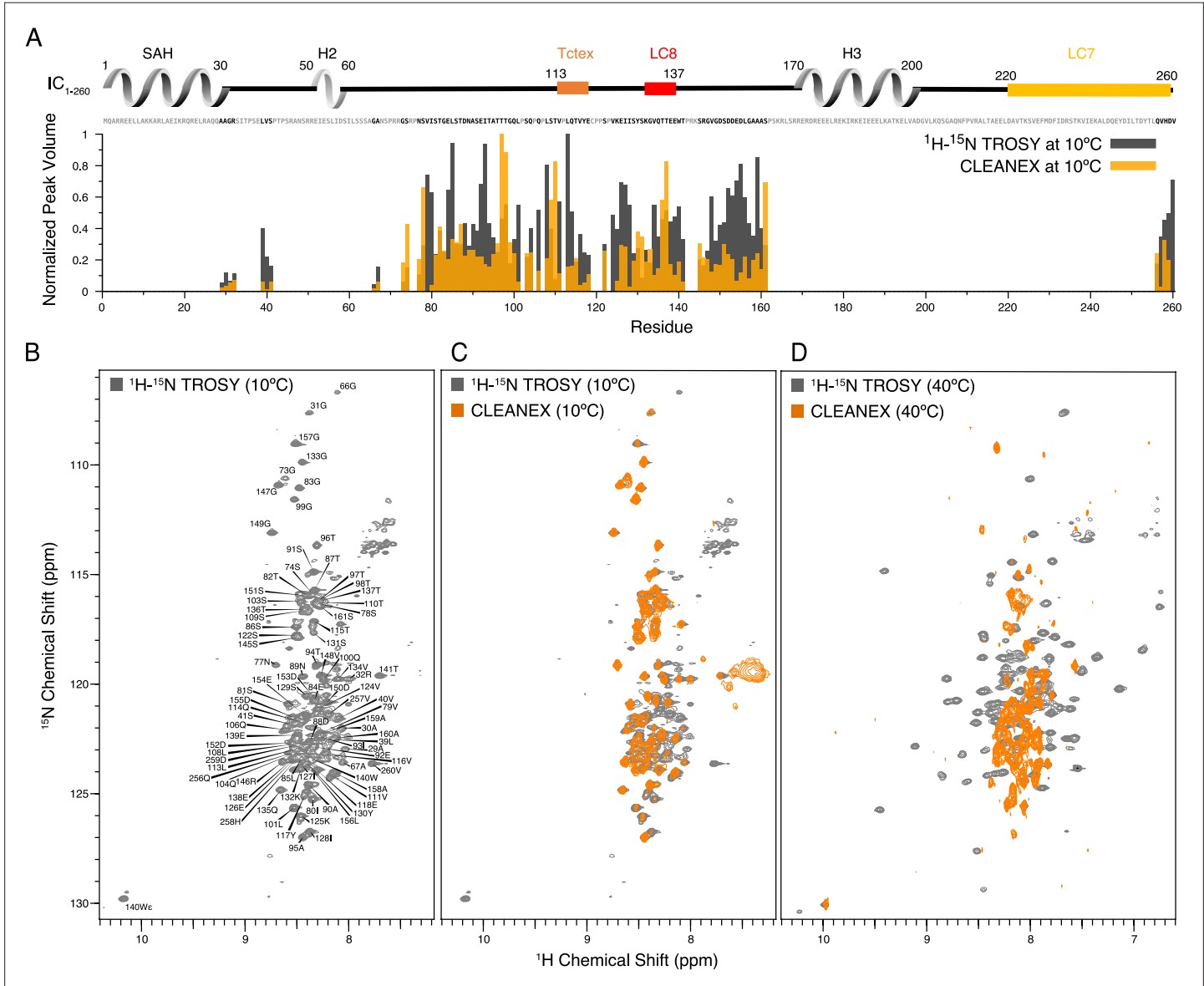

**Figure 4.** Identification of disordered linkers of *Chaetomium thermophilum* IC$_{1-260}$ using nuclear magnetic resonance spectroscopy. (**A**) Plot showing the normalized peak volumes at 10°C in the $^1$H-$^{15}$N TROSY spectrum (gray) and in the CLEANEX spectrum (orange) of the amides that could be assigned. Assigned residues are in black in the sequence above the plot (and unassigned residues in gray); all assigned residues are from disordered regions of IC$_{1-260}$. (**B**) $^1$H-$^{15}$N TROSY spectrum of IC$_{1-260}$ acquired at 800 MHz at 10°C showing amide assignments. (**C**) Overlay of a CLEANEX spectrum (orange) with the $^1$H-$^{15}$N TROSY spectrum (gray) at 10°C, shows that most of the assignable residues are in exchange with the solvent on the timescale of the CLEANEX experiment. (**D**) At 40°C, $^1$H-$^{15}$N TROSY spectrum (gray) shows new peaks appearing with greater chemical shift dispersion for IC$_{1-260}$ in the 800 MHz. Overlaying a CLEANEX spectrum (orange) at this temperature reveals that most of the new peaks in the $^1$H-$^{15}$N TROSY spectrum are from amides that are slow to exchange with the solvent and therefore are not observed in the CLEANEX spectrum.

The online version of this article includes the following source data and figure supplement(s) for figure 4:

**Source data 1.** Source files for IC$_{1-260}$ nuclear magnetic resonance (NMR) TROSY and CLEANEX data.

**Figure supplement 1.** Nuclear magnetic resonance spectra of intermediate chain (IC) are unaffected by salt concentration.

## Identifying disordered domains of Ct IC in the context of IC$_{1-260}$

To identify the disordered regions of IC$_{1-260}$, we used nuclear magnetic resonance (NMR) spectroscopy to collect $^1$H-$^{15}$N TROSY spectra as well as CLEANEX spectra, which measure rapid amide hydrogen exchange with water, of isotopically labeled protein samples (*Figure 4*). The limited chemical shift dispersion at 10°C (*Figure 4B*), along with appearance of the majority of the peaks in the CLEANEX experiment at this temperature (*Figure 4C*), indicates that the peaks observed in the spectra are for the disordered regions of IC$_{1-260}$. Using triple resonance experiments on a $^2$H/$^{13}$C/$^{15}$N-labeled sample, we assigned almost all of the observable peaks, corresponding to 37% (90 of 245) of the non-proline residues (*Figure 4B*). The handful of 'unassigned' peaks in *Figure 4B* mainly correspond to side-chain amides or peaks from minor conformers that could arise from cis/trans isomerization (*Alderson et al., 2018*). The assigned peaks all correspond to residues in disordered linker regions of IC$_{1-260}$, and these peaks vary considerably in intensity (*Figure 4A*).

Upon increasing the temperature to 40°C, peaks with much greater chemical shift dispersion become visible in the $^1$H-$^{15}$N TROSY spectrum (*Figure 4D*), while peaks corresponding to disordered regions of the protein disappear. Based on CD data (*Figure 3B*), IC$_{1-260}$ does not undergo significant secondary structural changes in the 10–40°C temperature range. Therefore, the newfound peak dispersion at higher temperature is not due to an increase in secondary structure. Rather, the appearance of peaks at elevated temperatures is most likely due to an increased rate of molecular tumbling causing peaks from more structured parts of the protein that were too broad to observe at lower temperatures to narrow and become visible. The conclusion that these peaks belong to residues in ordered regions is supported by their chemical shift dispersion and their absence in the CLEANEX spectrum at 40°C (*Figure 4D*). Some additional peaks are observed in the CLEANEX spectrum at 40°C that do not appear in the TROSY spectrum at this temperature; these correspond to amides in disordered regions of the protein that are in exchange with the solvent with exchange rates that makes them detectable in the CLEANEX experiment but invisible in the TROSY experiment. Together, these spectra support the conclusion that IC$_{1-260}$ contains both structured and disordered regions, with the most disordered regions resulting in peaks that are most easily assigned by NMR at low temperature. Even with the use of deuterated samples and TROSY-based experiments, the peaks at 40°C corresponding to more structured regions were, for the most part, not assignable as their rapid relaxation (potentially due to exchange broadening) resulted in extremely low signal intensities in backbone assignment experiments. IC$_{1-260}$ samples with salt concentrations of 20 mM and 250 mM were also explored at both 10 and 40°C, to ensure that electrostatic interactions were not the cause for missing peaks (*Figure 4—figure supplement 1*).

## The compact structure of Ct IC$_{1-260}$

To identify the residues involved in the tertiary contacts within IC$_{1-260}$, we collected NMR data on smaller fragments of IC: IC$_{1-88}$, IC$_{100-260}$, and IC$_{160-240}$ (*Figures 1 and 5A*). Based on the long-range interactions observed between IC$_{1-88}$ and IC$_{100-260}$ (*Figure 3B*, *Figure 3—figure supplement 2*), we hypothesized that an intramolecular interaction between the SAH region (residues 1–30) and the H3 region (residues 170–200) could be contributing to the stability of IC$_{1-260}$. The minimal peak disappearances and peak shifts for the spectrum of $^{15}$N-labeled IC$_{1-88}$ upon addition of unlabeled IC$_{160-240}$ indicate a limited degree of interaction (*Figure 5B–C*). However, when conducting the same experiment with unlabeled IC$_{100-260}$, the effects were much more pronounced (*Figure 5B–C*). In both cases, peaks corresponding to residues in the SAH and H2 regions of IC$_{1-88}$ either lost intensity or disappeared completely, while peaks from linker region residues remained largely unaffected, especially those that are near the C-terminus of IC$_{1-88}$. These data rule out a strong interaction between SAH and H3.

To narrow down the location of the interaction, an additional IC construct, IC$_{216-260}$, that does not include the H3 region, was $^{15}$N-labeled and completely assigned. Upon titration with $^{15}$N-labeled IC$_{1-88}$, many of the IC$_{216-260}$ peaks disappear or shift (*Figure 5D–E*). From this, we conclude that the tertiary contacts within IC$_{1-260}$ are largely between the SAH/H2 regions and a region that overlaps with the LC7 binding site. We note that these interactions appear to be even stronger in the context of the longer disordered chain encompassing the Tctex and LC8 binding sites based on the more substantial disappearances of peaks from the SAH/H2 regions when $^{15}$N-labeled IC$_{1-88}$ was titrated with unlabeled IC$_{100-260}$ (*Figure 5C*).

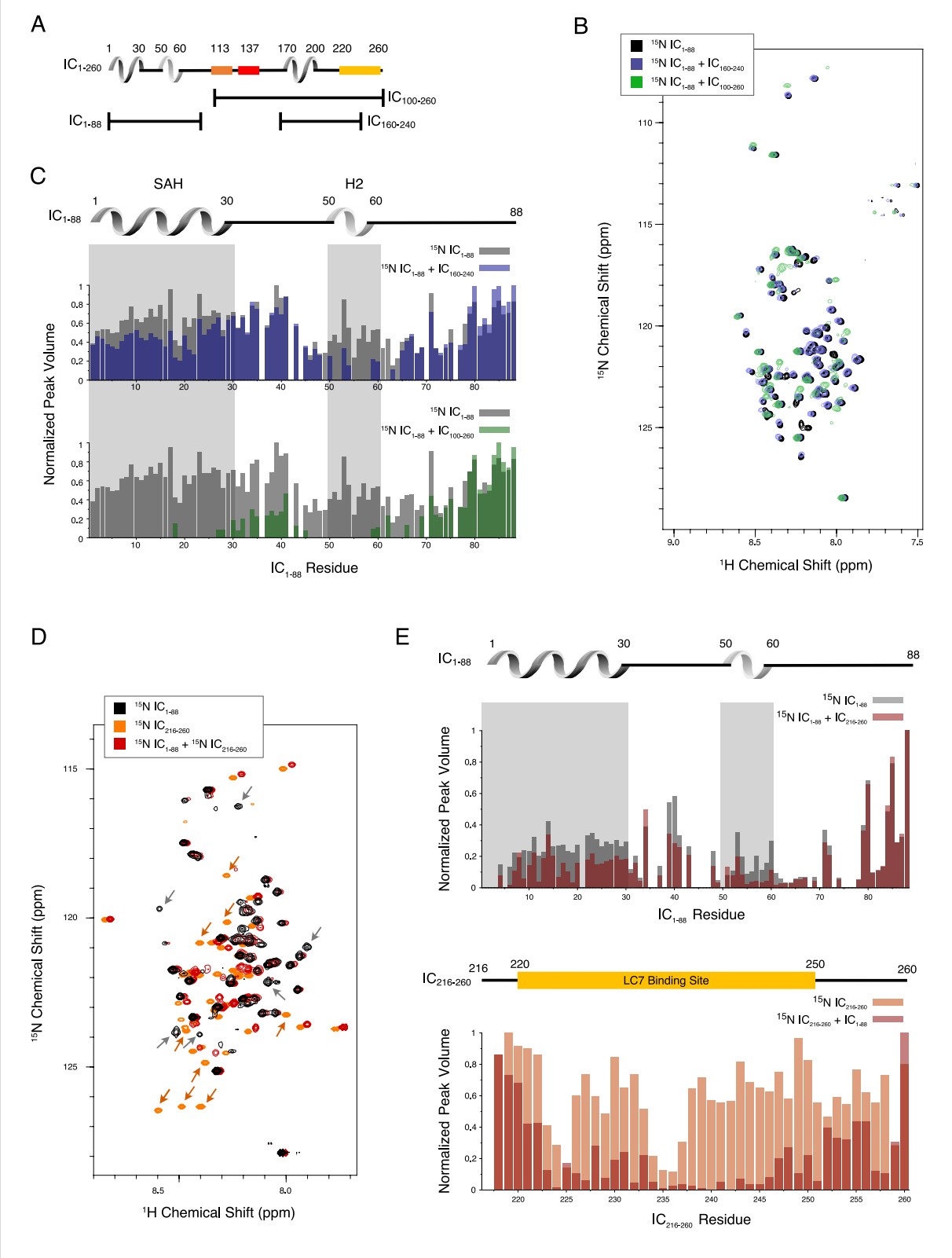

**Figure 5.** Evidence of tertiary contacts between the N and C-termini within *Chaetomium thermophilum* IC$_{1-260}$. (**A**) Domain architecture diagram for IC$_{1-260}$ with bars shown below corresponding to the IC$_{100-260}$, IC$_{1-88}$, and IC$_{160-240}$ constructs. (**B**) $^1$H-$^{15}$N TROSY overlays of free $^{15}$N-labeled IC$_{1-88}$ (black) and $^{15}$N-labeled IC$_{1-260}$ bound to unlabeled IC$_{160-240}$ (purple) and IC$_{100-260}$ (green). Note, spectra are deliberately offset in the $^1$H dimension to help visualize overlapping peaks. (**C**) Normalized peak volumes in the $^1$H-$^{15}$N TROSY spectra for (top) $^{15}$N-labeled IC$_{1-88}$ (gray) and $^{15}$N-labeled IC$_{1-88}$ + IC$_{160-240}$ (purple)

*Figure 5 continued on next page*

*Figure 5 continued*

and (bottom) $^{15}$N-labeled IC$_{1-88}$ (gray) and $^{15}$N-labeled IC$_{1-88}$ + IC$_{100-260}$ (green). (**D**) $^1$H-$^{15}$N HSQC overlay of $^{15}$N-labeled IC$_{1-88}$ (black), $^{15}$N-labeled IC$_{216-260}$ (orange), and $^{15}$N-labeled IC$_{216-260}$ bound to $^{15}$N-labeled IC$_{1-88}$ (red). Arrows highlight some of the more significant peak disappearances for IC$_{1-88}$ (gray arrows) and IC$_{216-260}$ (orange arrows). Note, spectra are deliberately offset by 0.03 ppm in the $^1$H dimension to help visualize overlapping peaks. (**E**) Normalized peak volumes in the $^1$H-$^{15}$N HSQC spectra for $^{15}$N-labeled IC$_{216-260}$ (top, gray columns) and $^{15}$N-labeled IC$_{216-260}$ (bottom, orange columns) when free and when in the presence of the other protein (red columns).

The online version of this article includes the following source data for figure 5:

**Source data 1.** Source files for nuclear magnetic resonance binding studies of intermediate chain (IC) fragments.

## Interactions of p150$_{CC1B}$ and NudE$_{CC}$ with Ct IC

The interaction between Ct IC$_{1-88}$ (which includes only the p150 and NudE binding domains) and p150$_{CC1B}$ has been previously reported (*Loening et al., 2020*) and is shown here only for comparative purposes (*Figure 6A and C*, top left). Here, we explore the interaction between IC$_{1-88}$ and NudE$_{CC}$ under similar conditions. By SV-AUC, IC$_{1-88}$ bound to p150$_{CC1B}$ shows a sedimentation coefficient of 3.7, whereas for IC$_{1-88}$ bound to NudE$_{CC}$ the coefficient is only 2.7 (*Figure 6A*). This is surprising considering that the two complexes have similar overall masses and binding stoichiometries and suggests that the complex with NudE$_{CC}$ is less compact than the complex with p150$_{CC1B}$. MALS data of the IC$_{1-88}$/p150$_{CC1B}$ complex gives a mass of 66 kDa, consistent with the mass expected for a p150$_{CC1B}$ dimer and two monomeric IC$_{1-88}$ chains (*Figure 6—figure supplement 1*) thus eliminating the possibility that the larger sedimentation coefficient can be explained by a larger than anticipated complex mass.

Previously published NMR titrations of unlabeled p150$_{CC1B}$ with $^{15}$N-labeled IC constructs that either contain the SAH region (IC$_{1-88}$) or do not (IC$_{37-88}$) demonstrate that both the SAH and H2 bind to p150$_{CC1B}$ (*Loening et al., 2020*; *Figure 6B*). When the same types of experiments were done with NudE$_{CC}$, however, peaks for residues corresponding to the H2 region did not disappear upon titration of $^{15}$N IC$_{37-88}$ with NudE$_{CC}$ (*Figure 6B*, top), indicating that the IC H2 region does not directly bind to NudE$_{CC}$ whereas this region does directly bind to p150$_{CC1B}$. This difference in binding is confirmed by ITC experiments using IC$_{37-88}$ that show binding with p150$_{CC1B}$ but no binding with NudE$_{CC}$ (*Figure 6—figure supplement 2*). For both p150$_{CC1B}$ and NudE$_{CC}$, titration into $^{15}$N IC$_{1-88}$ resulted in IC$_{1-88}$ peaks disappearing for both the SAH and H2 (*Figure 6B*, bottom). The disappearance of IC$_{1-88}$ peaks for both the SAH and H2 with NudE$_{CC}$ can be explained by weak engagement between NudE$_{CC}$ and H2 or an interaction between the SAH and H2, previously identified by NMR paramagnetic relaxation enhancement experiments (*Morgan et al., 2021*), that relays the change in correlation time of the SAH region upon binding to NudE$_{CC}$ to the H2 region.

From ITC measurements, we observe that, like IC$_{1-88}$ and p150$_{CC1B}$ (*Figure 6C*, top left), IC$_{1-88}$ and NudE$_{CC}$ bind with a 1:1 molar ratio (*Figure 6C*, bottom left), which corresponds to one NudE$_{CC}$ dimer binding two IC$_{1-88}$ monomeric chains. Data fit to a simple (single-step) binding model gave a dissociation constant ($K_d$) of 0.3 µM, which is weaker than the nM affinity estimated for IC$_{1-88}$ biphasic two-step binding to p150$_{CC1B}$ (*Loening et al., 2020*; *Supplementary file 1*). These differences in IC binding are conserved when a somewhat longer IC construct is used (IC$_{1-150}$, *Figure 6C*, center), whereas binding to p150$_{CC1B}$ and NudE$_{CC}$ is notably different or undetectable when using a construct containing the entire N-IC region (IC$_{1-260}$, *Figure 6C*, right). All p150$_{CC1B}$ isotherms were fit with a two-step binding model to provide the best possible fits, resulting in N values of 0.5, purposely imposed for longer IC constructs based on known overall 1:1 binding stoichiometry (*Supplementary file 1*; *Figure 6—figure supplement 1*). Further data and discussion of these observations are presented below.

## Interactions using Ct IC$_{1-260}$

SV-AUC was initially used to characterize complex formation between IC$_{1-260}$ and each of the other binding partners. The largest sedimentation coefficient was observed for the IC$_{1-260}$/LC8 complex; for all other complexes, a less dramatic peak shift was seen, an indicator of either weaker binding, a dynamic equilibrium between free and bound states, and/or a shift to a more elongated conformation (*Figure 7A*). For the IC$_{1-260}$/NudE$_{CC}$ complex, no significant change was observed in the SV-AUC data relative to unbound NudE$_{CC}$. However, the absence of a peak corresponding to free IC for the sample containing IC$_{1-260}$ and NudE$_{CC}$ indicates that some degree of binding takes place. This result along with

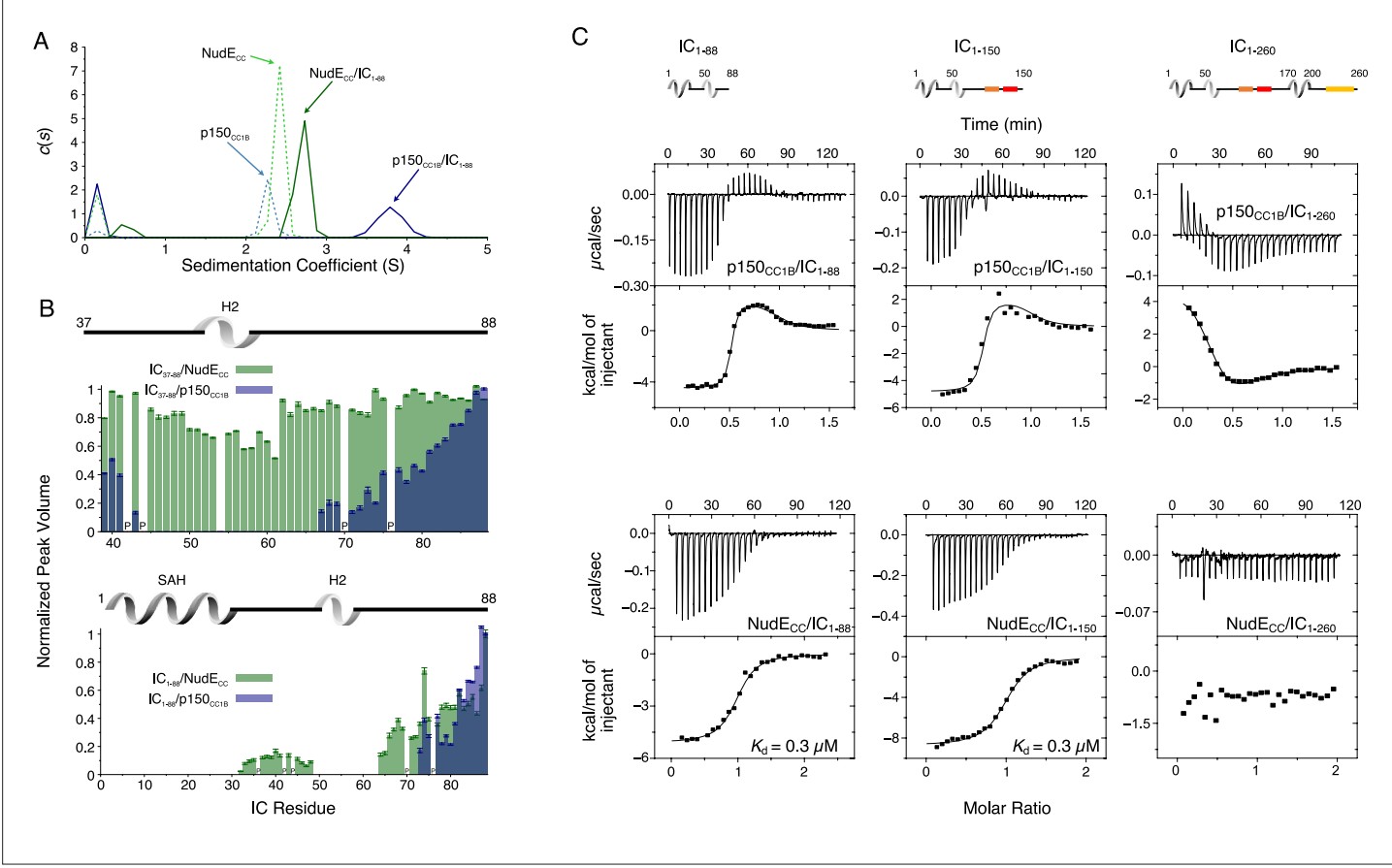

**Figure 6.** Binding interactions of *Chaetomium thermophilum* intermediate chain (IC) to p150$_{CC1B}$ and NudE$_{CC}$. (**A**) Sedimentation velocity analytical ultracentrifugation profiles for samples containing p150$_{CC1B}$ (blue dashed line), NudE$_{CC}$ (green dashed line), IC$_{1-88}$/p150$_{CC1B}$ complex (blue solid line), and IC$_{1-88}$/NudE$_{CC}$ complex (green solid line) show that IC$_{1-88}$ complexes have a larger sedimentation coefficient with p150$_{CC1B}$ than with NudE$_{CC}$. No data were collected for free IC$_{1-88}$ because it has no absorbance at 280 nm. (**B**) Normalized peak volumes in the $^1$H-$^{15}$N HSQC spectra for $^{15}$N-labeled IC$_{37-88}$ (top) or $^{15}$N-labeled IC$_{1-88}$ (bottom) when titrated with unlabeled p150$_{CC1B}$ (blue) and NudE$_{CC}$ (green). 'P' indicates proline residues. No peak disappearance for IC$_{37-88}$ was observed when NudE$_{CC}$ was added. Error bars are based on propagating the root-mean-square noise of the individual spectra. (**C**) Isothermal titration calorimetry (ITC) thermograms for p150$_{CC1B}$ titrated with IC$_{1-88}$ (top left), IC$_{1-150}$ (top middle), and IC$_{1-260}$ (top right), and for NudE$_{CC}$ titrated with IC$_{1-88}$ (bottom left), IC$_{1-150}$ (bottom middle), and IC$_{1-260}$ (bottom right), collected at 25°C (pH 7.5). Solid lines show fits to either a two-step binding model (p150$_{CC1B}$) or a one-site binding model (NudE$_{CC}$). For IC$_{1-260}$, reduced and endothermic binding is observed with p150$_{CC1B}$ whereas no binding is observed with NudE$_{CC}$.

The online version of this article includes the following source data and figure supplement(s) for figure 6:

**Source data 1.** Source files for sedimentation velocity analytical ultracentrifugation (SV-AUC) and nuclear magnetic resonance (NMR) binding studies of intermediate chain (IC) fragments with p150$_{CC1B}$ and NudE$_{CC}$.

**Figure supplement 1.** Size exclusion chromatography with multi-angle light scattering (SEC-MALS) of IC$_{1-88}$ and p150$_{CC1B}$.

**Figure supplement 1—source data 1.** This Excel workbook contains the data plotted for the size exclusion chromatography with multi-angle light scattering data shown in *Figure 6—figure supplement 1*.

**Figure supplement 2.** Binding interactions of *Chaetomium thermophilum* IC$_{37-88}$ to p150$_{CC1B}$ and NudE$_{CC}$.

our ITC (*Figure 6C*, bottom right) and NMR (*Figure 7B*, right) data for this complex indicates that binding of IC$_{1-260}$ to NudE$_{CC}$ is weak and is thus only detected at a high protein concentration.

Although we were unable to assign the NMR spectrum of IC$_{1-260}$ at 40°C because of unfavorable relaxation times, overlays of $^1$H-$^{15}$N TROSY spectra for free $^{15}$N-labeled IC$_{1-260}$ and binary complexes of $^{15}$N-labeled IC$_{1-260}$ with unlabeled binding partners (*Figure 7B*) show distinct patterns of peak disappearances. Upon addition of Tctex or LC8 (*Figure 7B*, left), only a handful of IC$_{1-260}$ peaks disappear and the patterns of disappearances are similar, as expected based on the proximity of the Tctex and LC8 binding sites. When either LC7, p150$_{CC1B}$, or NudE$_{CC}$ is added, considerably more peaks disappear

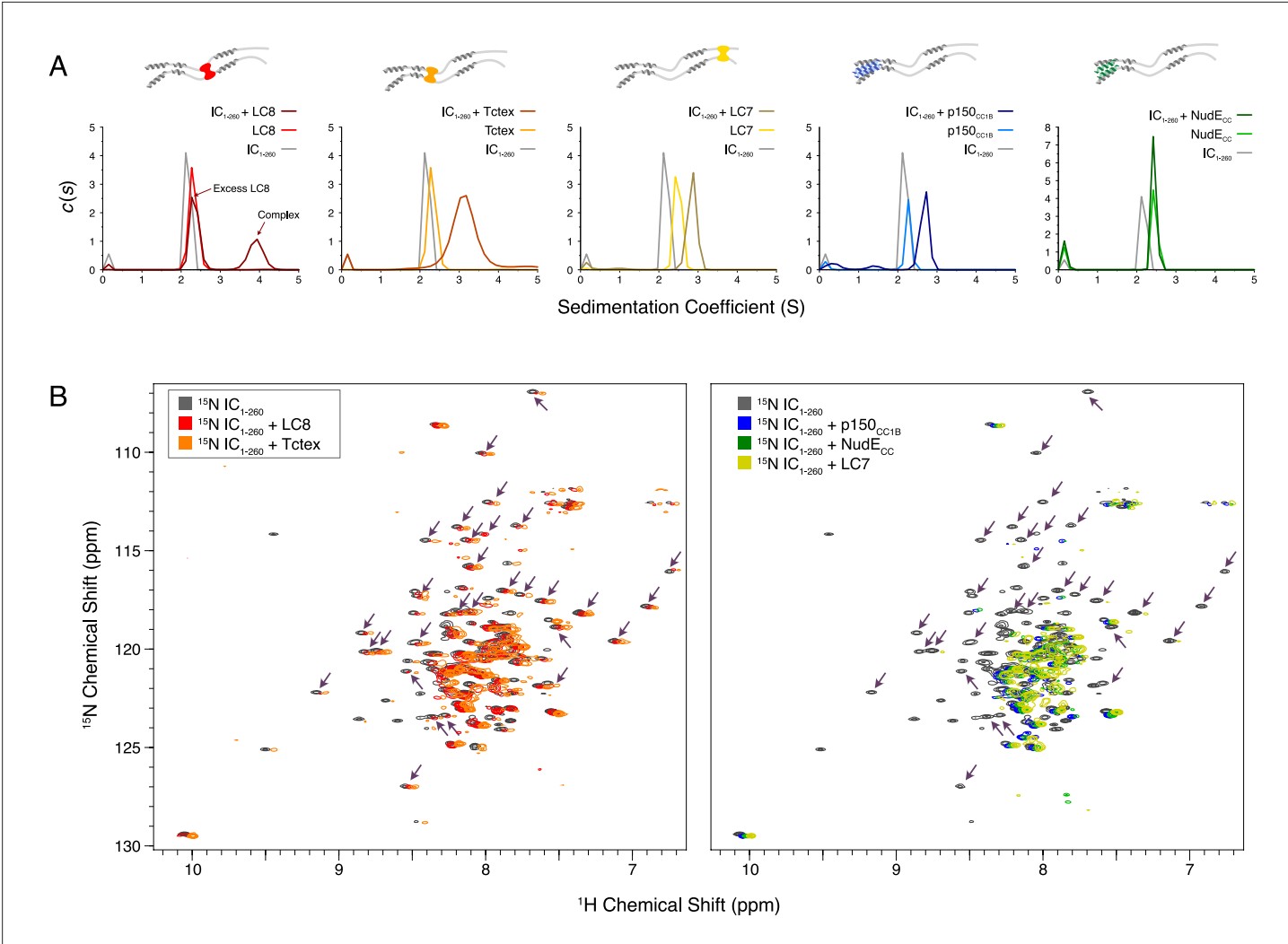

**Figure 7.** Binding characterization of binary complexes of $IC_{1-260}$. (**A**) Sedimentation velocity analytical ultracentrifugation of $IC_{1-260}$/LC8, $IC_{1-260}$/Tctex, $IC_{1-260}$/LC7, $IC_{1-260}$/p150$_{CC1B}$, and $IC_{1-260}$/NudE$_{CC}$. Data for the binary complexes is overlayed with data for each protein individually to better see shifts in the sedimentation coefficient of the binary complexes. (**B**) $^1$H-$^{15}$N TROSY overlays of free $^{15}$N-labeled $IC_{1-260}$ (black) and $^{15}$N-labeled $IC_{1-260}$ bound to unlabeled binding partners in a 1:1.5 molar ratio. The spectra were offset by 0.03 ppm in the $^1$H dimension to help illustrate changes in peak intensities. Changes in peak appearances/shifts/disappearances seem to be similar for LC8 (red) and Tctex (orange) versus changes seen for p150$_{CC1B}$ (blue), NudE$_{CC}$ (green), and LC7 (yellow). Arrows indicate peaks that remain when LC8 and Tctex are added to $IC_{1-260}$, but disappear when p150$_{CC1B}$, NudE$_{CC}$, and LC7 are added.

The online version of this article includes the following source data for figure 7:

**Source data 1.** Source files for sedimentation velocity analytical ultracentrifugation (SV-AUC) binding studies of $IC_{1-260}$.

in the spectra for $^{15}$N-labeled $IC_{1-260}$ (*Figure 7B*, right) and, surprisingly, similar patterns of disappearances occur even though the LC7 binding site is at the C-terminus of $IC_{1-260}$ whereas the p150$_{CC1B}$ and NudE$_{CC}$ binding sites are at the N-terminus. The similar patterns of peak disappearances suggest that regions of the N and C-termini of $IC_{1-260}$ interact in such a way that when one end of $IC_{1-260}$ is bound it affects the peak intensities of the other end and vice versa.

## Multivalency relieves IC autoinhibition

Following characterization of individual binding events, we sought to reconstitute full N-IC subcomplexes. Each subunit was first expressed and purified individually prior to dynein subcomplex formation ($IC_{1-260}$/Tctex/LC8/LC7), achieved by mixing $IC_{1-260}$ with the dynein light chains in a 1:1.5 (IC to LC) molar ratio. To this dynein subcomplex, p150$_{CC1B}$ or NudE$_{CC}$ was added to create two larger subcomplexes. Each subcomplex was re-purified by SEC to remove any excess of the binding partners (which

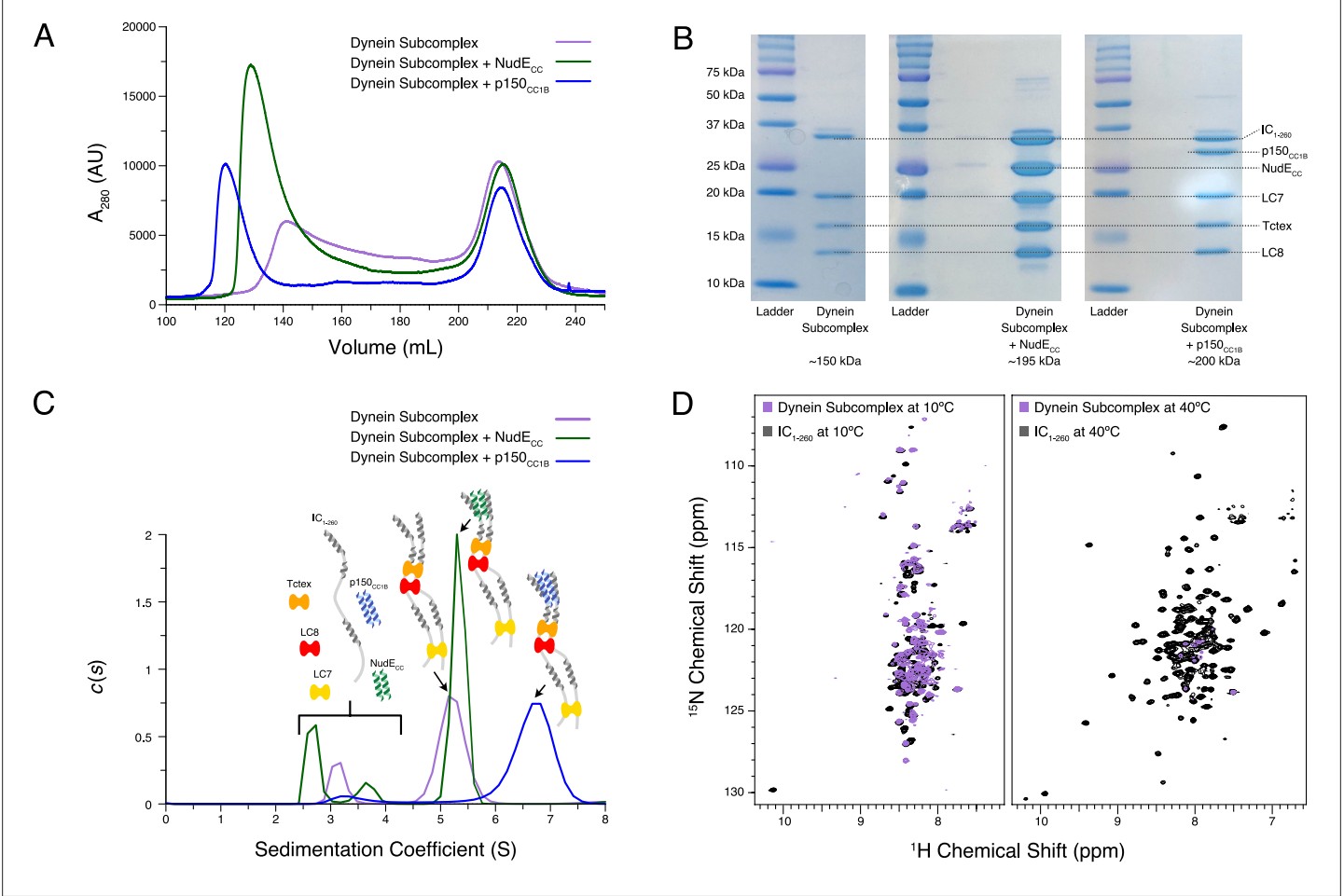

**Figure 8.** Reconstitution and characterization of dynein subcomplexes. (**A**) Size exclusion chromatography (SEC) traces of the dynein subcomplex (IC/light chains) (purple) and the dynein subcomplex with the addition of either p150$_{CC1B}$ (blue) or NudE$_{CC}$ (green). (**B**) Sodium dodecyl sulphate–polyacrylamide gel electrophoresis gels of fractions collected from SEC for all complexes showing all expected proteins. (**C**) SV-AUC profiles of the dynein subcomplex (purple) bound to p150$_{CC1B}$ (blue) or NudE$_{CC}$ (green). (**D**) $^1$H-$^{15}$N TROSY overlays of free IC$_{1-260}$ (black) and the dynein subcomplex (purple). At 10°C, many peaks are still of high intensity in the 153 kDa complex, indicating that some regions remain disordered. The very few peaks at 40°C of the bound are most likely due to the size and tumbling of the subcomplex and consistent with the fact that the majority of the peaks at this temperature are from ordered regions.

The online version of this article includes the following source data for figure 8:

**Source data 1.** Source files for size exclusion chromatography (SEC) and sedimentation velocity analytical ultracentrifugation (SV-AUC) of reconstituted IC$_{1-260}$ subcomplexes.

**Source data 2.** Original sodium dodecyl sulphate–polyacrylamide gel electrophoresis gel images.

elute at ~215 mL) and to assess their overall stability (**Figure 8A**). The shape and symmetry of the eluting SEC peak for the dynein subcomplex (~140 mL) indicate a weaker binding affinity than when either p150$_{CC1B}$ (~120 mL) or NudE$_{CC}$ (~130 mL) was added. In all cases, however, each expected subunit was detected by sodium dodecyl sulphate–polyacrylamide gel electrophoresis (SDS-PAGE) of collected fractions (**Figure 8B**), thus validating successful assembly.

Using SV-AUC, we show that when all dynein light and intermediate chains are present, the fully bound dynein subcomplex sediments as a single peak with a sedimentation coefficient of approximately 5 S (**Figure 8C**). Surprisingly, addition of NudE$_{CC}$, which adds approximately 45 kDa of mass, only slightly increases the sedimentation coefficient. In contrast, addition of p150$_{CC1B}$ increases the sedimentation coefficient to approximately 7 S. The difference in the SV-AUC data between the addition of p150$_{CC1B}$ and NudE$_{CC}$ is surprising based on the similar expected masses of the complexes (201 and 198 kDa, respectively). A difference in overall complex mass due to a binding stoichiometry

greater than 1:1 for p150$_{CC1B}$ and IC could explain the complex's larger sedimentation coefficient; however, this is very unlikely. As shown in previous studies across multiple species, p150$_{CC1B}$ consistently binds IC in a 1:1 stoichiometry (*Morgan et al., 2011*; *Morgan et al., 2021*; *Jie et al., 2017*; *Nyarko et al., 2012*; *Loening et al., 2020*). Furthermore, visualization of the complexes by SDS-PAGE (*Figure 8B*) shows similar intensity ratios between IC$_{1-260}$/NudE$_{CC}$ and IC$_{1-260}$/p150$_{CC1B}$, indicating that they exist in the same stoichiometry. Calculated frictional coefficient $f/f_0$ ratios ($f_0$ is the frictional coefficient of a theoretical sphere of the same mass and partial specific volume) of the dynein subcomplex alone, with p150$_{CC1B}$, and with NudE$_{CC}$ are 2.03, 1.75, and 2.38, respectively. These values indicate that the subcomplex with p150$_{CC1B}$ is more globular/spherical than the other subcomplexes. For comparison, IDPs and elongated proteins usually have $f/f_0$ ratios over 2, whereas globular proteins typically have a ratio of 1.2 (*Edwards et al., 2020*). We propose that the observed difference in the $f/f_0$ ratios between the p150$_{CC1B}$ and the NudE$_{CC}$ subcomplexes is due to differences in their overall shapes, details of which need to be further examined. In earlier work on *Drosophila* N-IC (*Morgan et al., 2021*), binding with NudE shifts the equilibrium of IC toward more open and elongated states, so it is reasonable to assume a similar effect occurs with Ct IC. Also, important to note is that the shift in sedimentation coefficient for p150$_{CC1B}$ when added to the IC/light chains complex is significantly more pronounced than that with IC$_{1-260}$ alone, suggesting that binding to IC$_{1-260}$ is significantly enhanced in the presence of the light chains.

NMR spectroscopy indicates that most of the peaks in the disordered regions observed at 10°C for free IC$_{1-260}$ (*Figure 8D*) remain visible in the spectrum of the dynein subcomplex (IC$_{1-260}$/Tctex/LC8/LC7), indicating that there is still significant disorder in the fully bound complex (153 kDa). In contrast, comparison of the spectra of free IC$_{1-260}$ and dynein subcomplex at 40°C shows a drastic disappearance of peaks, indicating that the residues corresponding to these peaks have much longer correlation times in the bound state either due to their involvement in binding or due to the overall increase in weight of the entire subcomplex. Peaks from linker regions not involved in binding light chains that are observed at 10°C but not at 40°C likely disappear due to rapid exchange with the solvent at the higher temperature.

## Effect of light chains on autoinhibition is retained in full-length Ct IC

To determine if the autoinhibition seen in IC$_{1-260}$ is present in the full-length construct and thus biologically relevant, IC$_{FL}$ (residues 1–642) was produced using a baculovirus expression system. The estimated mass of 75.4 kDa for IC$_{FL}$ from SEC-MALS matches closely to the expected monomeric mass of 79 kDa (*Figure 9A–B*). Furthermore, SV-AUC shows a single, homogenous peak with a sedimentation coefficient of 4.0 S, as would be expected for monomeric IC$_{FL}$ (*Figure 9B* top). This is the first evidence that full-length IC is a monomer in solution and requires the light chains for its dimerization. Adding either p150$_{CC1B}$ or NudE$_{CC}$ to IC$_{FL}$ results in a negligible shift of the IC$_{FL}$ peak and, in both cases, show a peak corresponding to unbound p150$_{CC1B}$ or NudE$_{CC}$ (*Figure 9B*). This lack of binding between IC$_{FL}$ and p150$_{CC1B}$/NudE$_{CC}$ shows that autoinhibition occurs in IC$_{FL}$ in a similar manner to what we have observed in IC$_{1-260}$. SV-AUC experiments on the dynein subcomplex show that IC$_{FL}$ bound to the three light chains (IC$_{FL}$/Tctex/LC8/LC7) has a sedimentation coefficient of 7.0 S but shifts to values of 7.5 and 8 S upon addition of NudE$_{CC}$ and p150$_{CC1B}$, respectively (*Figure 9B* middle). These results mimic those seen for IC$_{1-260}$ and confirm that the addition of the light chains allows p150$_{CC1B}$ or NudE$_{CC}$ to bind by relieving IC autoinhibition.

To further explore the role of each of the light chains in relieving IC autoinhibition, SV-AUC experiments with IC$_{FL}$/Tctex/LC8 and IC$_{FL}$/LC7 were conducted. The IC$_{FL}$/Tctex/LC8 and IC$_{FL}$/LC7 complexes show peaks with sedimentation coefficients of approximately 5.2 and 4.2 S, respectively (*Figure 9B*). Interestingly, the IC$_{FL}$/Tctex/LC8 complex exhibits no shift upon the addition of NudE$_{CC}$ but does shift to 6.2 S upon the addition of p150$_{CC1B}$ (*Figure 9B*). The IC$_{FL}$/LC7 complex, on the other hand, exhibits a shift upon addition of either NudE$_{CC}$ or p150$_{CC1B}$ to 4.5 or 5 S, respectively (*Figure 9B*). These data suggest that the addition of LC7 is sufficient to allow binding of p150$_{CC1B}$ and NudE$_{CC}$ to IC$_{FL}$, whereas the binding of Tctex and LC8 only promotes p150$_{CC1B}$ binding to IC$_{FL}$ and not NudE$_{CC}$.

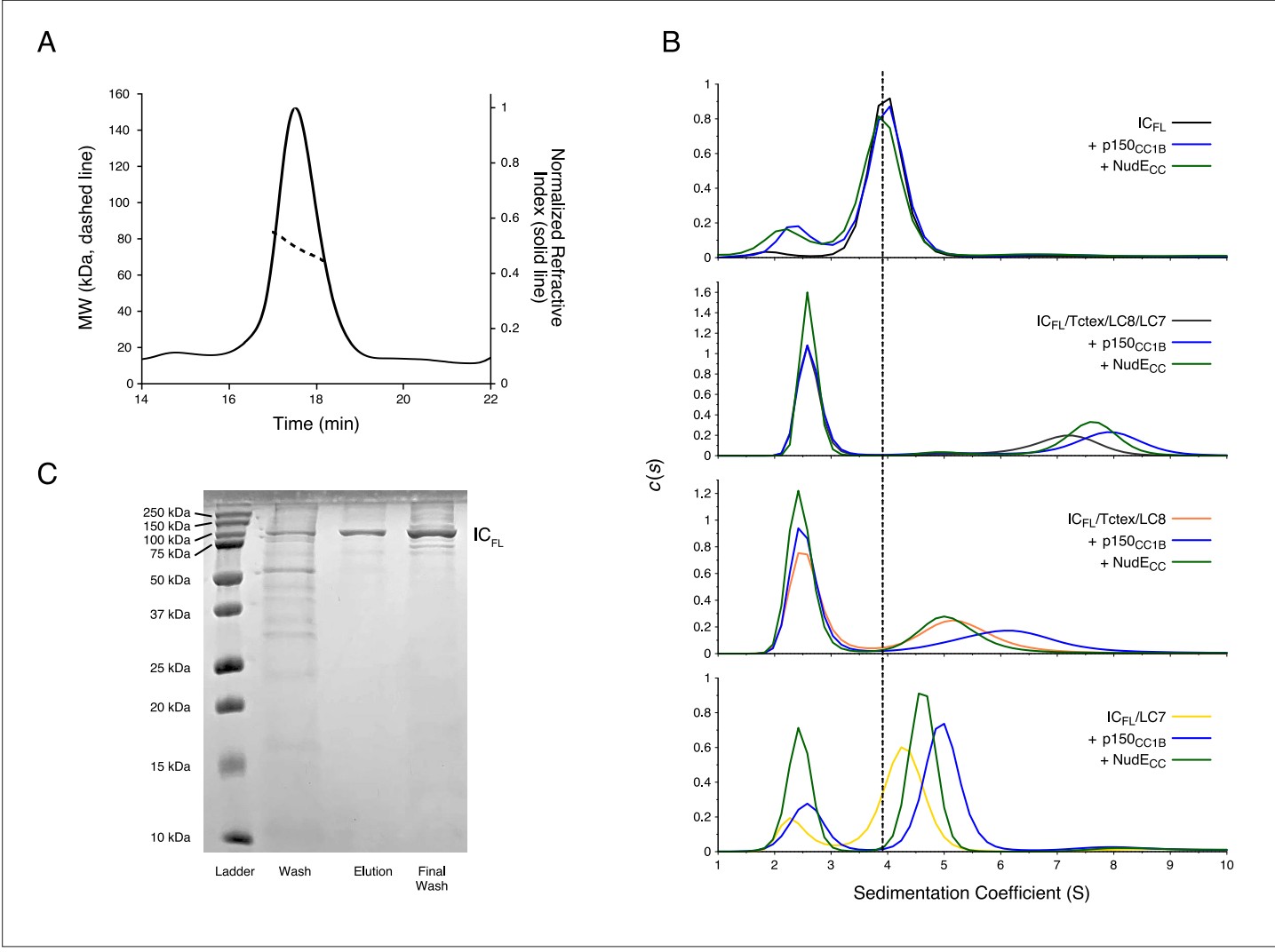

**Figure 9.** Binding characterization of *Chaetomium thermophilum* IC$_{FL}$ subcomplexes. (**A**) The estimated mass of IC$_{FL}$ from multi-angle light scattering is 75.4 kDa, which indicates that IC$_{FL}$ is a monomer in the absence of binding partners. (**B**) Sedimentation velocity analytical ultracentrifugation profiles of IC$_{FL}$ (black), IC$_{FL}$ mixed with p150$_{CC1B}$ (blue) or NudE$_{CC}$ (green), and the subcomplexes: IC$_{FL}$/Tctex/LC8/LC7 (gray), IC$_{FL}$/Tctex/LC8/LC7/p150$_{CC1B}$ (blue), IC$_{FL}$/Tctex/LC8/LC7/NudE$_{CC}$ (green), IC$_{FL}$/Tctex/LC8 (orange), IC$_{FL}$/Tctex/LC8/p150$_{CC1B}$ (blue), IC$_{FL}$/Tctex/LC8/NudE$_{CC}$ (green), IC$_{FL}$/LC7 (yellow), IC$_{FL}$/LC7/p150$_{CC1B}$ (blue), and IC$_{FL}$/LC7/NudE$_{CC}$ (green). The black, dashed line is centered on unbound IC$_{FL}$ to help guide the eye. (**C**) Sodium dodecyl sulphate–polyacrylamide gel electrophoresis gel of immobilized metal affinity chromatography fractions (left to right: wash, elution, and final wash) with a band for IC$_{FL}$ migrating in accordance with the expected mass of ~79 kDa.

The online version of this article includes the following source data for figure 9:

**Source data 1.** Source files for size exclusion chromatography with multi-angle light scattering (SEC-MALS) and sedimentation velocity analytical ultracentrifugation (SV-AUC) of IC$_{FL}$.

**Source data 2.** Original sodium dodecyl sulphate–polyacrylamide gel electrophoresis gel image.

## Discussion

N-IC from a variety of species contains a stretch of about 300 amino acids that are primarily disordered, except for a few short α-helices. Within the first 40 residues is a fully ordered helix (the SAH region), followed by a short disordered linker and another region (H2) that form a fully ordered helix in some species but are only a nascent helix in others (*Jie et al., 2015*; *Jie et al., 2017*; *Loening et al., 2020*; *Siglin et al., 2013*). Prior work suggests that a more disordered H2 is correlated with tighter IC/p150$^{Glued}$ binding and has shown that, for Ct IC, the H2 region is disordered and binds directly to p150$^{Glued}$ (although mostly in a nonspecific manner) (*Loening et al., 2020*). In this work, we use a construct of IC that encompasses almost its entire N-terminal domain to probe the interactions of Ct

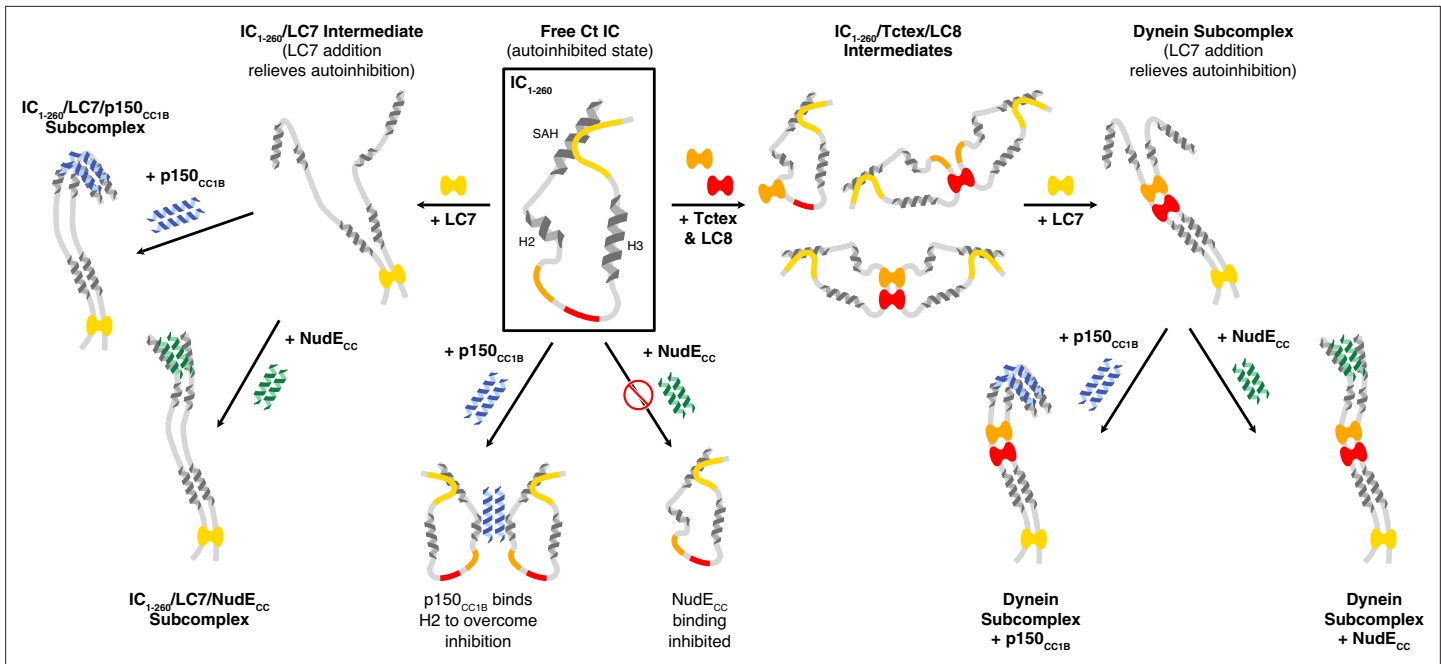

**Figure 10.** A model of *Chaetomium thermophilum* IC$_{1-260}$ binding interactions and subcomplex assemblies. Free IC$_{1-260}$ is compact and in autoinhibited state (Boxed in black), with the single α-helix (SAH), H2, and H3 regions depicted as helices and with colors indicating the LC8 (red), Tctex (orange), and LC7 (yellow) binding sites. When LC7 is added (left arrow) LC7 outcompetes autoinhibition to bind IC$_{1-260}$, exposing the SAH domain for p150$_{CC1B}$ and NudE binding. When p150$_{CC1B}$ or NudE$_{CC}$ is added (down arrows) to free IC$_{1-260}$, autoinhibition prevents NudE$_{CC}$ from binding and reduces the binding affinity of p150$_{CC1B}$. However, as p150$_{CC1B}$ is able to bind to the H2 region of IC$_{1-260}$, binding is not completely prevented. Addition of Tctex and/or LC8 (right arrow) leads to a number of possible binary and ternary intermediates. LC8's role of driving IC dimerization is depicted and, in all intermediates, we predict that IC autoinhibition remains based on very limited changes in nuclear magnetic resonance spectra. Continuing to the right, the addition of LC7 leads to formation of the dynein subcomplex and the release of SAH autoinhibition. The free SAH is now able to resume transient interactions with H2 prior to binding with either p150$_{CC1B}$ or NudE$_{CC}$. Finally, addition of p150$_{CC1B}$ or NudE$_{CC}$ leads to the formation of the p150 and NudE subcomplexes (bottom right). Our sedimentation velocity analytical ultracentrifugation data suggest that the NudE subcomplex adopts a more elongated conformation.

IC with both p150$_{CC1B}$ and NudE$_{CC}$ in context of its multivalent assembly with the three dimeric dynein light chains. Furthermore, we demonstrate for the first time, using full-length IC, that the mechanisms at place in IC$_{1-260}$ remain in the context of the entire IC protein. The results are illustrated by a model that explains the different aspects of IC autoinhibition and how the assembly of the multivalent IC subcomplex relieves this autoinhibition (*Figure 10*).

## Ct IC$_{1-260}$ is a partially disordered compact monomer

The primary Ct IC construct (IC$_{1-260}$) used in this work is, to date, the longest IC construct from any species made recombinantly and extensively studied by NMR, ITC, and SV-AUC (*Morgan et al., 2011*; *Jie et al., 2015*; *Hall et al., 2010*; *Hall et al., 2009*; *Jie et al., 2017*; *Nyarko et al., 2012*; *Wainman et al., 2009*; *Liang et al., 2007*; *Siglin et al., 2013*). IC$_{1-260}$ far exceeds previously studied constructs from Ct (res. 1–35, 37–88, and 1–88) as well as constructs from *Drosophila* (res. 1–60, 30–143, 84–143, 1–143, 92–260, and 114–260) and rat (res. 1–44, 1–96, and 1–112). Similar to IC from *Drosophila*, free Ct IC$_{1-260}$ is monomeric (*Figure 2A and D*) and contains significant disorder (*Figure 4*). IC$_{1-260}$ resists unfolding at temperatures of up to 50°C (*Figure 3B*), a feature that we attribute to long-range tertiary contacts between the N- and C-termini (*Figure 5*, *Figure 3—figure supplement 2*). SV-AUC and SEC-MALS experiments confirm the presence of a binding interaction between IC$_{1-88}$ and IC$_{100-260}$ (*Figure 3—figure supplement 2*), and NMR titrations with N- and C-terminal IC constructs identify an intramolecular interaction between residues in the SAH and the LC7 binding site of IC$_{1-260}$ (*Figure 5*). These long-range tertiary interactions impact IC binding interactions, as seen by the dramatic reduction in binding of p150$_{CC1B}$ or NudE$_{CC}$ with IC$_{1-260}$ compared to IC$_{1-88}$ and IC$_{1-150}$ (*Figure 6C*).

## Different modes of binding of IC to p150$^{Glued}$ and NudE

The coiled-coil domains of p150$^{Glued}$ and NudE (p150$_{CC1B}$ residues 478–680, and NudE$_{CC}$ residues 1–190) show similar dimeric structure and stability by AUC and CD (*Figure 2*). However, despite their many similarities, p150$_{CC1B}$ and NudE$_{CC}$ have different modes of binding to Ct IC. First, IC interactions with p150$_{CC1B}$ involve binding to both the SAH and H2 regions of IC (*Loening et al., 2020*). In contrast, NudE$_{CC}$ binds IC via the SAH region and not the H2 (*Figure 6B*, *Figure 6—figure supplement 2*), agreeing with results previously reported for constructs from *Drosophila* (*Nyarko et al., 2012*). Second, the p150$_{CC1B}$/IC$_{1-88}$ complex has a larger sedimentation coefficient when compared to the NudE$_{CC}$/IC$_{1-88}$ complex (3.8 S vs 2.7 S, *Figure 6A*) even though the complexes have similar expected masses and the same binding stoichiometry. This suggests that the p150$_{CC1B}$/IC$_{1-88}$ complex has a more compact structure. Third, although binding is much weaker with IC$_{1-260}$ (*Figure 6C*) compared to the smaller IC constructs for both p150$_{CC1B}$ and NudE$_{CC}$, binding to NudE$_{CC}$ is significantly weaker and is only detected at the higher concentrations and enhanced sensitivity afforded by NMR experiments (*Figure 7B*). Finally, similar to the short IC constructs, the light chain/IC assembled complex shows a different sedimentation coefficient when bound to p150$_{CC1B}$ (~7 S) or NudE$_{CC}$ (~5 S) (*Figure 8C*). The larger sedimentation coefficient for the p150$_{CC1B}$ subcomplex is unexplained by the small mass difference and thus indicates a more compact and stable conformation for the subcomplex when p150$_{CC1B}$ is bound. SV-AUC experiments with IC$_{FL}$ further confirm that p150$_{CC1B}$ and NudE$_{CC}$ have different binding modes as, once again, complexes with assembled IC$_{FL}$/light chain subcomplexes show a larger sedimentation coefficient when bound to p150$_{CC1B}$ (~8 S) than when bound to NudE$_{CC}$ (~7.5 S) (*Figure 9*). We propose that binding of NudE$_{CC}$ induces a more elongated and open conformation of the IC subcomplex while p150$_{CC1B}$ binding results in a more compact conformation as depicted in the model in *Figure 10*.

The advantages of a compact or elongated IC subcomplex remain to be elucidated. It is possible that these different conformations aid in dynein functions that are linked to binding either dynactin (cargo recruitment, processivity, and spindle formation) or NudE (dynein recruitment to kinetochores and membranes, centrosome migration, spindle orientation, and LIS1 interactions). Additionally, variations in IC subcomplex conformation may impact other regulatory mechanisms, such as phosphorylation at residues located in the serine-rich linker following H2 in mammalian IC isoforms (*Jie et al., 2017*; *Vaughan et al., 2001*). Regulatory impacts and the significance of different mechanisms may also rely heavily on linker lengths that vary between isoforms and such variability in linker length may have unequal contributions to IC autoinhibition.

## Autoinhibition in IC selects for binding of p150$^{Glued}$ over NudE

Interactions of IC$_{1-260}$ with its binding partners observed by ITC and AUC show weak endothermic binding to p150$_{CC1B}$ but no binding to NudE$_{CC}$. Our data suggest a process, which we refer to as autoinhibition, in which IC$_{1-260}$ adopts a compact structure that covers the SAH region necessary for binding to NudE$_{CC}$ and for strong binding to p150$_{CC1B}$, but still leaves the H2 region partly accessible so that weak binding to p150$_{CC1B}$ can still occur (*Figure 10*). The SAH region is made inaccessible by tertiary interactions within IC, as seen in an NMR titration of $^{15}$N-labeled IC$_{1-88}$ with IC$_{100-260}$ (*Figure 5B–C*). We assign these tertiary interactions to those within the LC7 binding site at the C-terminus via NMR titrations of $^{15}$N-labeled IC$_{216-260}$ with IC$_{1-88}$ (*Figure 5D–E*), but note that the binding affinity does appear stronger in the longer construct (IC$_{100-260}$). This increase in affinity could be due to additional interactions between H2 and H3 (*Figure 3—figure supplement 2*, *Figure 5B–C*), and indicates that the full context of the disordered chain, including the binding sites for Tctex, LC8, and LC7, is needed for strong, intramolecular interactions. These interactions likely have an autoinhibitory effect, which would explain the reduced binding affinity for p150$_{CC1B}$ and NudE$_{CC}$ for IC$_{1-260}$ in comparison to the shorter constructs. A need for 'opening' IC from an autoinhibitory state prior to binding also explains the entropically driven first binding step observed by ITC, especially for p150$_{CC1B}$/IC binding (*Supplementary file 1*). Since the inhibitory effect is more pronounced for IC$_{1-260}$ binding with NudE$_{CC}$, than for IC$_{1-260}$ binding with p150$_{CC1B}$, we propose autoinhibition as a mechanism for selection of p150$_{CC1B}$ over NudE$_{CC}$. Autoinhibition as a selection mechanism is underscored by our results with full-length IC (*Figure 9*). Although binding to both p150$_{CC1B}$ and NudE$_{CC}$ is inhibited with IC$_{FL}$ alone, the addition of Tctex and LC8 rescues binding to p150$_{CC1B}$, likely by helping make the nearby H2 region more accessible. In contrast, binding to both p150$_{CC1B}$ and NudE$_{CC}$ is rescued via LC7 binding to IC$_{FL}$ (*Figure 9*).

With light chains present and autoinhibition relieved, competition between NudE$_{CC}$ and p150$_{CC1B}$ may result from the ability of p150 to invade an IC/NudE complex by first binding the exposed H2 region before competing for the SAH binding site as previously observed (**Nyarko et al., 2012**).

Our work establishes the SAH region as a critical structural element of the IC that, as a single helix, packs against other helices stabilizing a compact structure and, in the process, inhibits binding of partners to either the SAH region or to the partner helix it packs against. The SAH/LC7 site intramolecular contacts, for example, inhibit IC from interacting with p150$_{CC1B}$ and NudE$_{CC}$. Based on its inherent ability to interact with other helices, the SAH could form long-range interactions with other helices in the dynein motor or dynein adaptors and inhibit other interactions. This is a highly likely possibility since autoinhibition is a common regulatory mechanism for dynein as is observed in interactions with the motor domain and dynein activators dynactin and Bicaudal-D1 (**Zhang et al., 2017**; **Torisawa et al., 2014**; **Tripathy et al., 2014**; **Terawaki et al., 2015**; **Kobayashi et al., 2017**).

### Autoinhibition in IC regulation emphasizes interplay of disorder and multivalency

Recombinant expression and biophysical characterization of multiple dynein and non-dynein subunits and their reconstitution into three assembled IC subcomplexes and characterization of these complexes offer a testable model for the interplay of disorder and multivalency in regulation of dynein function and selection of binding partners.

The N-terminal domain and full-length IC constructs from Ct demonstrate subcomplex assembly via light chain binding as the primary modulating mechanism for p150$^{Glued}$ and NudE binding. This is an effect that can only be detected with larger constructs of IC, such as Ct IC$_{1-260}$, that contain long disordered linkers separating short helices and binding sites. As summarized by the model shown in **Figure 10**, we propose that these properties combine to create a multifaceted system of regulation for IC. The disordered linkers allow for the needed flexibility to bring together the SAH region and the C-terminus (200+ amino acids away) and to extend it into an open conformation when LC7 is bound (either on its own, or with the other two light chains), which suggests a unique role for LC7 in regulating IC interactions. The model highlights that the C-terminal end of IC$_{1-260}$, which contains a portion of the LC7 binding site, interacts with the SAH and results in a closed conformation. The closed conformation is autoinhibitory, preventing binding of the SAH region by p150$_{CC1B}$ or NudE$_{CC}$. However, because p150$_{CC1B}$ also binds to H2, p150$_{CC1B}$ binding to IC$_{1-260}$ is not completely abolished. Our data (**Figure 5E**) suggest that residues that overlap with part of the LC7 binding site make the strongest contacts with the SAH, leaving flanking residues to initiate binding to LC7 (**Figures 5 and 9**). From the limited changes in NMR spectra upon Tctex and LC8 binding, we conclude that the overall closed conformation of IC is likely retained by binding of LC8 and/or Tctex since they bind in the center of a long linker, whereas LC7 binding is required to release the SAH region, thereby allowing assembled IC to fully bind to p150$_{CC1B}$ and NudE$_{CC}$. In future work it will be important to demonstrate that autoinhibition and its subsequent reversal by multivalent dynein light chain binding is a conserved process of IC among species and determine how this autoinhibition affects dynein function.

## Materials and methods

**Key resources table**

| Reagent type (species) or resource | Designation | Source or reference | Identifiers | Additional information |
|---|---|---|---|---|
| Gene (*Chaetomium thermophilum*) | Ct IC | NA | UniProt: G0SCF1-1 | |
| Strain, strain background (*Escherichia coli*) | Rosetta (DE3) | Sigma-Aldrich | Catalog number: 70954 | |
| Cell line (Insect) | Sf9 | ThermoFisher | Catalog number: 11496015 | Used within 1 yr of purchase |
| Strain, strain background (*E. coli*) | DH10EMBacY | Geneva Biotech | DH10EMBacY | Used within 1 yr of purchase |
| Peptide, recombinant protein | Ct IC full length protein | GenScript | | pFastbac1 vector |
| Peptide, recombinant protein | IC$_{1-88}$ | Ref. 32 | | |

*Continued on next page*

Continued

| Reagent type (species) or resource | Designation | Source or reference | Identifiers | Additional information |
|---|---|---|---|---|
| Peptide, recombinant protein | IC$_{37-88}$ | GenScript | | |
| Peptide, recombinant protein | IC$_{160-240}$ | GenScript | | |
| Peptide, recombinant protein | IC$_{216-260}$ | Azenta Life Sciences | | |
| Chemical compound, drug | Ammonium- $^{15}$N chloride | Sigma Aldrich | Catalog number: 299251 | |
| Chemical compound, drug | D-glucose $^{13}$C$_6$ | Sigma Aldrich | Catalog number: 389374 | |
| Software, algorithm | Origin 7.0 | OriginLab | | |
| Software, algorithm | SEDFIT | Open-source | | |
| Software, algorithm | TopSpin 3.6 | Bruker Biospin Corporation | RRID:SCR_014227 | |
| Software, algorithm | CcpNmr analysis (CCPN) | Ref. 80 | RRID:SCR_016984 | |
| Software, algorithm | NMRPipe | Ref. 78 | | |

## Cloning, protein expression, and purification

All studies were carried out using constructs from Ct (thermophilic fungus, G0SCF1-1). IC$_{1-260}$ (res. 1–260), p150$_{CC1B}$ (res. 478–680), and NudE$_{CC}$ (res. 1–190) constructs were prepared by PCR and cloned into a pET-24d vector with an N-terminal 6× His tag using the Gibson Assembly protocol (*Gibson et al., 2009*; *Gibson et al., 2010*). In addition, a fragment of the IC$_{1-260}$ construct with N-terminal residues removed (IC$_{100-260}$) was generated by the same method. IC$_{37-88}$ and IC$_{160-240}$ were ordered from GenScript (Piscataway, NJ), and IC$_{216-260}$ was ordered from Azenta Life Sciences (Chelmsford, MA). Full-length Ct Tctex, LC8, and LC7 were amplified out of a Ct cDNA library and cloned into a pET-15b vector with an uncleavable C-terminal 6× His tag. These light chain constructs contain a single, non-native GS linker prior to the tag sequence. For IC, p150, and NudE constructs, an N-terminal tobacco etch virus (TEV) protease cleavage site was included to allow the removal of the 6× His tag, leaving a non-native GAH sequence post cleavage. DNA sequences were verified by Sanger sequencing. The IC$_{1-88}$ construct was prepared previously (*Loening et al., 2020*).

Recombinant plasmids were transformed into Rosetta (DE3) *Escherichia coli* cells (Merck KGaA, Darmstadt, Germany) for protein expression. Bacterial cultures for expression of unlabeled proteins were grown in ZYM-5052 autoinduction media at 37°C for 24h (*Studier, 2005*), whereas cultures for expression of isotopically labeled ($^{15}$N or $^{15}$N/$^{13}$C) proteins were grown in MJ9 minimal media (*Jansson et al., 1996*) at 37°C to an OD$_{600}$ of 0.8 before being induced with 0.4-mM isopropyl-β-D-1-thiogalactopyranoside and continuing expression overnight at 26°C. Proteins were purified from the cell cultures by immobilized metal affinity chromatography (IMAC) using previously published methods (*Loening et al., 2020*). Complete cleavage of the tag by TEV protease was verified by SDS-PAGE analysis.

Proteins were further purified using a Superdex 75 (Cytiva Life Sciences, Pittsburgh, PA) SEC column and then, for IC$_{1-260}$ samples, followed by anion exchange using Macro-Prep High Q Support resin (Bio-Rad, Hercules, California) with elution in 0.1–0.2 M sodium chloride. Protein concentrations were determined from absorbance at 205 and 280 nm (*Anthis and Clore, 2013*). Molar extinction coefficients for the constructs used are as follows ($\varepsilon_{205}$ and $\varepsilon_{280}$): IC$_{1-260}$ = 853,230 and 11,460 M$^{-1}$ cm$^{-1}$, IC$_{1-150}$ = 488,290 and 8480 M$^{-1}$ cm$^{-1}$, IC$_{100-260}$ = 665,620 and 14,440 M$^{-1}$ cm$^{-1}$, IC$_{160-240}$ = 387,110 and 2980 M$^{-1}$ cm$^{-1}$, p150$_{CC1B}$ = 685,820 and 9970 M$^{-1}$ cm$^{-1}$, NudE$_{CC}$ = 665,800 and 16,960 M$^{-1}$ cm$^{-1}$, Tctex = 649,850 and 31,970 M$^{-1}$ cm$^{-1}$, LC8=495,640 and 8480 M$^{-1}$ cm$^{-1}$, and LC7=533,510 and 6990 M$^{-1}$ cm$^{-1}$. All purified proteins were stored at 4°C with a protease inhibitor mixture of pepstatin A and phenyl-methanesulfonyl fluoride and used within 1 week.

## Full-length IC cloning and expression

Full-length Ct IC (IC$_{FL}$) was expressed in Insect Sf9 cells (Thermo Fisher, used within 1 year of purchase). The sequence of IC$_{FL}$ was codon optimized and cloned into pFastbac1 vector by Genscript (Piscataway, NJ) with an N-terminal 6× His tag followed by a TEV protease cleavage site. IC$_{FL}$ was expressed in Sf9 cells using the multiBAC system following previously published protocols with slight modifications

(*Zhang et al., 2017*; *Schlager et al., 2014*). The plasmid was transformed into DH10EmBacY cells (free of mycoplasma contamination) (Geneva Biotech, used within 1 year of purchase) and single, white colonies were selected after 2 days and inoculated into 2× TY media supplemented with antibiotics and grown overnight. The bacteria pellet was harvested at 4000 rpm for 10 min and resuspended in 0.3 mL QIAGEN miniprep buffer P1, followed by 0.3 mL P2 buffer. After 5 min incubation, 0.4 mL P3 buffer was added, and the mixture was centrifuged at 13,000 rpm for 10 min. The supernatant was added to 0.8-mL ice cold isopropanol, and bacmid DNA was pelleted for 10 min at 13,000 rpm. The pellet was washed twice with 70% ethanol, resuspended in $H_2O$, and stored at 4°C.

Sf9 cells (free of mycoplasma contamination) were cultured in SF-900 III serum-free media (Thermo Fisher) at a shaking speed of 125 rpm at 27°C. 2 µg fresh bacmid DNA was transfected into 2 mL Sf9 cells at $0.9 \times 10^6$ cells/2 mL with 8 µL Cellfection II (Thermo Fisher) following manufacturer's protocol. Five days later, 0.5 mL of the transfected culture medium was added to a 50 mL culture of Sf9 cells ($0.5 \times 10^6$ cells/mL) for P2 infection. Four days later, cells were spun down at 3000 rpm for 15 min at 4°C, and the supernatant of P2 virus was collected and stored at 4°C in the dark. Protein expression was induced by adding P2 virus to Sf9 cells (1:100 ratio, $2 \times 10^6$ cells/mL). After 4 days, cells were harvested by centrifugation at 4000 rpm for 20 min at 4°C. The pellet was flash frozen in liquid nitrogen and stored at −80°C for further protein purification. Affinity purification was carried out as described above for other constructs.

### IC structure prediction

Sequences for IC from a range of species were obtained from the UniProt protein database (*Consortium, 2019*). *Rattus norvegicus* (rat, UniProt: Q62871-3), *Drosophila melanogaster* (fruit fly, UniProt: Q24246-11), *Saccharomyces cerevisiae* (yeast, UniProt: P40960-1), *Homo sapiens* (human, UniProt: O14576-2), *Danio rerio* (zebrafish,UniProt: A1A5Y4-1), *Callorhinchus milii* (Australian ghost shark, UniProt:V9KAN3-1), *Octopus bimaculoides* (Californian two-spot octopus, UniProt:A0A0L8HM30-1), *Caenorhabditis elegans* (nematode, UniProt: O45087-1), and Ct (thermophilic fungus, UniProt: G0SCF1-1). The first 260 amino acids of the IC from each species were scored using the Agadir algorithm, which outputs a prediction for percent helicity per residue (*Muñoz and Serrano, 1997*; *Muñoz and Serrano, 1994*; *Muñoz and Serrano, 1995a*; *Muñoz and Serrano, 1995b*; ). Agadir is not developed for prediction of long protein sequences, therefore results were compared to predictions from PSIPRED (*Buchan and Jones, 2019*; *Nucleic Acids Research, 2021*).

### Circular dichroism

CD measurements were made using a JASCO (Easton, Maryland) J-720 CD spectropolarimeter. Samples consisted of proteins at concentrations of 5–10 µM in 10-mM sodium phosphate buffer (pH 7.5). All experiments were done using a 400-µL cuvette with a path length of 0.1 cm. The data shown are the average of three scans. Thermal unfolding data were collected in increments of 5°C over a temperature range of 5–60°C. CD measurements were used to estimate the fractional helicity of the samples using the equation below:

$$\text{Fractional Helicity} = \frac{\left(\theta_{222}^{\text{exp}} + 3000\ \text{deg cm}^2\ \text{dmol}^{-1}\right)}{-36{,}500\ \text{deg cm}^2\ \text{dmol}^{-1}}$$

where $\theta_{222}^{\text{exp}}$ is the experimentally observed residue ellipticity (MRE) at 222 nm (*Wei et al., 2014*).

### Isothermal titration calorimetry

ITC experiments were conducted using a MicroCal VP-ITC microcalorimeter (Malvern Panalytical, United Kingdom). All experiments were performed at 25°C and with protein samples in a buffer containing 50-mM sodium phosphate (pH 7.5), 50-mM sodium chloride, and 1-mM sodium azide. Samples were degassed at 25°C prior to loading. Each experiment was started with a 2 µL injection, followed by 27–33 injections of 10 µL. Protein concentrations in the cell ranged from 20 to 40 µM, and concentrations in the syringe ranged from 200 to 400 µM. The data were processed using Origin 7.0 (Malvern Panalytical, Malvern, UK) and fit to either a single-site or a two-site binding model. The recorded data (*Supplementary file 1*) are the averages of two to three independent experiments.

## NMR measurements and analysis

Samples for NMR were prepared in a 20-mM sodium chloride, 50-mM phosphate (pH 7.4) buffer that included 5% $D_2O$, 1-mM sodium azide, 0.2-mM 2,2-dimethylsilapentane-5-sulfonic acid (DSS), and 1× cOmplete protease inhibitor cocktail (Roche, Basel, Switzerland). NMR spectra were collected over a temperature range of 10–40°C using a Avance III HD 800 MHz spectrometer (Bruker Biospin, Billerica, Massachusetts) with a TCI cryoprobe and a Avance NEO 600 MHz spectrometer (Bruker Biospin) equipped with a room-temperature TXI probe. Band-selective excitation short transient variants of TROSY-based triple resonance sequences (HNCO, HNCA, HN(CO)CA, HN(CA)CO, HNCACB, and HN(CO)CACB) were used for backbone assignment of $IC_{1-260}$ at 10°C (*Solyom et al., 2013*). Assignments of $^{15}N$-labeled $IC_{216-260}$ were carried out using $^{15}N$-seperated TOCSY and NOESY experiments. The accessibility of $IC_{1-260}$ amide protons to exchange with the solvent was determined by measuring peak volumes in a Fast HSQC spectrum with a 20-ms CLEANEX-PM mixing period. The $IC_{1-260}$ and $IC_{216-260}$ concentrations were 350–500 µM for these samples.

NMR experiments of binary complexes (IC with one binding partner) were performed by combining $^{15}N$-labeled $IC_{1-260}$ with unlabeled $p150_{CC1B}$, $NudE_{CC}$, Tctex, LC8, or LC7 at a molar ratio of 1:1.5. For the $IC_{216-260}$/$IC_{216-260}$ binary complex, the $^{15}N$-labeled components were mixed in a 1:1 molar ratio and each was at a concentration of 350 µM. Spectra for each binary complex were collected at both 10 and 40°C. NMR experiments of the dynein subcomplex ($IC_{1-260}$ with all light chains) were performed much in the same way. The $IC_{1-260}$ concentration was 250 µM for these samples. NMR data were processed using TopSpin 3.6 (Bruker) and NMRPipe (*Delaglio et al., 1995*). For 3D experiments that employed non-uniform sampling the spectra were reconstructed using SCRUB (*Coggins et al., 2012*). Peak assignment was performed using CCPN analysis 2.5.2 (*Vranken et al., 2005*). For all plots of peak volumes (*Figures 4A, 5C, E and 6B*), the peak volumes for each spectrum were normalized to the volume for the largest peak in that spectrum.

## Analytical ultracentrifugation

SV-AUC experiments were performed using a Beckman Coulter Optima XL-A analytical ultracentrifuge, equipped with absorbance optics. For individual proteins, the concentration used was 15–30 µM. For binary complexes, $IC_{1-260}$ or $IC_{FL}$ was mixed with each binding partner at ratios of 1:1.5 or 1:2 (molar ratio of IC to binding partner). The buffer condition used for all SV-AUC experiments was 25-mM tris(hydroxymethyl)aminomethane hydrochloride (pH 7.4), 150-mM KCl, 5-mM tris (2-carboxyethyl) phosphine, and 1-mM sodium azide. Samples were loaded into standard, 12-mm path length, two-channel sectored centerpieces and centrifuged at 42,000 rpm and 20°C. 300 scans were acquired at 280–297 nm with no interscan delay. Data were fit to a $c(s)$ distribution using SEDFIT (*Schuck, 2000*). Frictional coefficients were calculated based on determined S values and expected molecular weights.

SE-AUC experiments were performed on the same instrument. $NudE_{CC}$ and $p150_{CC1B}$ were each loaded with three concentrations in the range of 15–60 µM in six channel centerpieces, centrifuged at three speeds (10,000, 14,000, and 18,000 rpm) and scanned at 280 nm. Samples were scanned every 3 h until they were at equilibrium (i.e. when the final sequential scans were superimposable), which occurred after 30–36 h of centrifugation. Data were acquired as averages of five measurements of absorbance at each radial position, with a nominal spacing of 0.003 cm between each position. The data from the three speeds and three concentrations were globally fit to a monomer-dimer self-association model and resulted in random residuals. Other models tested did not give adequate variances and random residuals. All experiments were done at 20°C. Data were fit using HETERO-ANALYSIS (*Help, 2021*).

## SEC multiangle light scattering

SEC-MALS was carried out using a Superdex 200 gel filtration column on an AKTA fast liquid chromatography system (Cytiva Life Sciences) coupled with a DAWN MALS detector and an Optilab refractive index detector (Wyatt Technology, Santa Barbara, CA). Data for $IC_{1-260}$ was collected for protein samples at a concentration of 200 µM protein in a buffer composed of 50-mM sodium phosphate (pH 7.5), 50-mM sodium chloride, and 1-mM sodium azide. Data for $IC_{FL}$ was collected for protein samples at a concentration of 30 µM in a buffer composed of 25-mM tris(hydroxymethyl)aminomethane hydrochloride (pH 7.4), 150-mM KCl, 5-mM β-mercaptoethanol, and 1-mM sodium azide. Molar mass and

error analysis were determined using ASTRA v9, employing a Zimm light scattering model (Wyatt Technology).

## Subcomplex reconstitution

Post purification of individual protein, $IC_{1-260}$ was combined with dynein light chains (Tctex, LC8, and LC7). To the assembled subcomplex, the non-dynein proteins $p150_{CC1B}$ or $NudE_{CC}$ were added in a 1:1.5 molar ratio prior to SEC using a Superdex 200 column (Cytiva Life Sciences). Subcomplex reconstitution was verified by SDS-PAGE of SEC fractions, as each protein clearly resolves. When estimating concentrations for complexes purified by SEC, the absorbance at 280 nm was used with the assumption that the majority of formed complex in solution followed the expected stoichiometry of 1:1 (IC monomer:partner monomers) and that very little excess of free protein would be present.

## Acknowledgements

We acknowledge support from the National Science Foundation (Award 1617019 for E.J.B. and Award 2003557 for N.M.L.). The Oregon State University NMR Facility is funded in part by the National Institutes of Health (HEI Grant 1S10OD018518) and by the M J Murdock Charitable Trust (Grant 2014162). The Lewis & Clark College Bruker Avance NEO 600 NMR spectrometer was purchased with support from the National Science Foundation (Award 1917696) and the M J Murdock Charitable Trust (Grant 201811283).

## Additional information

### Funding

| Funder | Grant reference number | Author |
| --- | --- | --- |
| National Science Foundation | 1617019 | Elisar J Barbar |
| National Science Foundation | 2003557 | Nikolaus M Loening |
| National Institute of Biological Resources | 1S10OD018518 | Elisar J Barbar |

The funders had no role in study design, data collection and interpretation, or the decision to submit the work for publication.

### Author contributions

Kayla A Jara, Conceptualization, Formal analysis, Investigation, Methodology, Writing - original draft; Nikolaus M Loening, Formal analysis, Funding acquisition, Investigation, Methodology, Writing - review and editing; Patrick N Reardon, Formal analysis, Investigation, Methodology, Writing - review and editing; Zhen Yu, Data curation, Investigation, Methodology; Prajna Woonnimani, Coban Brooks, Cat H Vesely, Investigation; Elisar J Barbar, Conceptualization, Resources, Supervision, Funding acquisition, Project administration, Writing - review and editing

### Author ORCIDs

Kayla A Jara http://orcid.org/0000-0003-0406-2957
Nikolaus M Loening http://orcid.org/0000-0002-5074-6906
Elisar J Barbar http://orcid.org/0000-0003-4892-5259

### Decision letter and Author response

Decision letter https://doi.org/10.7554/eLife.80217.sa1
Author response https://doi.org/10.7554/eLife.80217.sa2

## Additional files

### Supplementary files

• Supplementary file 1. ITC results for intermediate chain interactions with p150$_{CC1B}$ and NudE$_{CC}$ at 25°C. [a]For the p150$_{CC1B}$ experiments, a two-site binding model was used as it provided much better fits when compared to a single site model. [b]Fitting of this experiment was done with set $N$ values due to the amount of extra noise in the data and thus the $N$ values are reported without error. [c]This experiment did not show evidence of binding and thus the data was not fit to a binding model.

• MDAR checklist

### Data availability

Source data is included.

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
