## [Editor Report]

This is a scholarly paper. The identification of an auto-inhibitory mechanism of the dynein intermediate chain provides an important contribution to our understanding of dynein assembly. It further illustrates the plethora of regulatory mechanisms attainable by IDPs and the range of different biophysical techniques that can be used to elucidate them.

---

## [Decision Letter]

**Decision letter after peer review:**

Thank you for submitting your article "Multivalency, autoinhibition, and protein disorder in the regulation of interactions of dynein intermediate chain with dynactin and the nuclear distribution protein" for consideration by *eLife*. Your article has been reviewed by 3 peer reviewers, and the evaluation has been overseen by a Reviewing Editor and Volker Dötsch as the Senior Editor. The reviewers have opted to remain anonymous.

The reviewers do mention that some additional attention should be placed on clarifying the biological significance of the system with a better introduction (including graphics). Further, in the Discussion section more emphasis on the biological relevance of the findings should be given; for example what is the biological role of autoinhibition. As you can see some of the reviewers ask for additional experiments to clarify aspects of the paper. Please consider these carefully in a revision.

Essential revisions:

1) The authors propose that the two-step binding isotherm observed for p150 is due to binding to both the SAH and H2 regions of IC(1-88), while NudE shows a single binding event by ITC due to interaction with the SAH region only. The NMR experiments of IC(1-88) do not provide sufficient support for this hypothesis (Figure 6B, bottom panel; see point 8 below as well). Additional experimental data would be needed to fully support this conclusion.

2) ITC, NMR, and AUC data are presented for the binding of NudE to IC(1-260). Some more clarification is needed in terms of the interpretation of these data, also in the context of the observations on IC(FL). The experimental observations do not seem to be explainable simply by a weak complex or a concentration-dependent effect as suggested by the authors.

3) The difference in sedimentation coefficient of the dynein subcomplex + NudE and of the dynein subcomplex + p150 is surprisingly large suggesting significantly different shapes of the two bound complexes. Some discussion of this issue is present in the manuscript, but no clear explanation is provided. It would seem necessary to confirm these observations by other complementary techniques. With respect to the SV-AUC data, it would be helpful to use frictional coefficients f/F0 to disentangle Mw and conformation from sedimentation coefficients.

4) The authors suggest that the binding of LC7 releases the auto-inhibitory interaction of IC, however, NMR does not directly support this conclusion (Figure 7B). Some discussion of why this long-range interaction inhibits the binding of NudE, but not LC7 itself, should be included.

5) As indicated in the summary above more care should be placed on introducing the system. For example, it was felt that Figure 1B, did not help much (e.g. the light chains were hard to acknowledge as they appear to be rather small compared to the IC chain and were at first overlooked at just the binding sites; where is the heavy chain of dynein, why is there no coiled coil of p150, etc?). In a related manner, the functional role of autoinhibition should be more clearly emphasized in the discussion, e.g., what is the relevance of the differential binding of the two ligands and their differential effects on the autoinhibited state. Which biological outcomes are to be expected?

6) One of the conclusions is that the internal contacts occur between the C-terminal of the IC and SAH/H2, which is seen from the intensity changes in the HSQC spectra upon addition of the 160-240 construct to the 1-88 construct. However, adding the linker part from 100-160 produces a much more pronounced effect (Figure 5C, bottom), suggesting that residues in this region, which includes the Tctex and LC8 binding motifs play additional roles. Is the binding of the light chains to IC of higher affinity to the 100-160 protein than to the 1-260? In that case, this could suggest that also inhibitory access to these two sites occurs in the autoinhibited state. The additional effect of the 100-160 residues should be addressed.

7) Can H3 be excluded as a player in the internal interactions, just because you see binding to the LC7 site when studied in isolation? Once the LC7 regions is bound, H3 may also participate, as also clearly indicated from the data shown in Figure 4A. Using an H3 peptide would be relevant.

8) Fig6B and associated text: there is a clear although weak loss in intensity/peak volume in the H2 region for the interaction with NudE. Why assume that there is no interaction? The affinity for NudE is lower, so the concentration of the complex will also be lower at similar conditions compared to that of p150, and this would give rise to the smaller effects in the spectra. In the lower panel, there is a clear indication of binding to H2 as well, and SAH and H2 binding may very well be cooperative as they are sequentially close. What are the relative concentrations of NudE and p150 in the cell? Would they be competitive despite the difference in affinities? Can a mechanism for p150 ability to relieve autoinhibition be proposed – from Fig6B, could it be able to bind first to the H2 region even though SAH is involved in the autoinhibitory interaction?

[Editors' note: further revisions were suggested prior to acceptance, as described below.]

Thank you for resubmitting your work entitled "Multivalency, autoinhibition, and protein disorder in the regulation of interactions of dynein intermediate chain with dynactin and the nuclear distribution protein" for further consideration by *eLife*. Your revised article has been evaluated by Volker Dötsch (Senior Editor) and a Reviewing Editor.

The manuscript has been improved but there are some remaining issues that need to be addressed, as outlined below:

*Reviewer #1 (Recommendations for the authors):*

I have received the revised version of the manuscript by Jara et al. Overall, the authors have done a good job in the revision process, in particular by adding new data to support their conclusions. I am overall in favour of publishing the manuscript in *eLife*.

One point where I still remain less convinced is about the origin of the lack of direct correlation between the ITC, NMR and AUC data. The authors suggest that these techniques are differentially sensitive to protein concentration and to the presence of minor/open and auto-inhibited states of IC. This could be true; however, only a little evidence for this is actually provided.

*Reviewer #2 (Recommendations for the authors):*

The authors have carefully considered and addressed the review comments, and the manuscript is ready for publication in my opinion.

*Reviewer #3 (Recommendations for the authors):*

The authors have thoroughly revised the manuscript and included further evidence to support their conclusion, and the relevance of the data for the biology of the protein complex is now addressed to some extent in the discussion.

A few comments remain:

Figure 1 has improved but can be further improved by adding N – and C to the IC ends, indicate the HC (heavy chain), LIC (dynein light intermediate chain), N-IC (N-terminal domain of IC), C-IC, SAH (single α helix), H2 (short helix), all of which are mentioned in the text. Perhaps use a zoom on the region central to the study. Also, the use of CT-IC is confusing, and the authors should use more correct CtIC to indicate the reference to the species.

Although I agree with the authors that the interaction with NudE is dominated by the SAH interaction, some interaction with H2 is seen in the data in Figure 6b, with some intensity changes for isolated H2 binding, and loss of signals for H2 in the 1-88 construct, and with the linker residues between SAH and H2 still detectable. This adds to the explanation of the larger binding affinity in the longer construct. The weak engagement of H2 with NudE may allow p150 to invade a NudE:IC complex, possibly in the trimeric complex (see also work by Berlow et al. (Nature 2017 and follow up paper)) and may be of biological relevance for regulation. This offers an alternative interpretation of the data, p. 12, l. 268-271, also relevant for discussion p. 17, l. 405-406, and p. 417.

The auto-inhibited state may be further stabilized. Data in Figure 5 suggest interaction between SAH and the LC7 (as noted by the authors), but also additional interaction between H2 and H3 (4C-top compared to 4C bottom), not mentioned.

I appreciate the inclusion of the ITC data, and the two-state binding fit, but the data is not discussed. With N-values of 0.5 and a dominantly entropically driven interaction, how is this interpreted by the authors.

---

## [Author Response]

Essential revisions:1) The authors propose that the two-step binding isotherm observed for p150 is due to binding to both the SAH and H2 regions of IC(1-88), while NudE shows a single binding event by ITC due to interaction with the SAH region only. The NMR experiments of IC(1-88) do not provide sufficient support for this hypothesis (Figure 6B, bottom panel; see point 8 below as well). Additional experimental data would be needed to fully support this conclusion.

We agree that we currently do not have sufficient support for the hypothesis that p150_CC1B_ exhibits two-step binding with IC due to discrete binding events of the SAH and H2, and thus have edited the manuscript to exclude this language. Instead, we emphasize the conclusion that p150_CC1B_ binds to both the SAH and H2 regions of IC and that the SAH is needed for tight binding while NudE_CC_ binds strongly only to the SAH. We have provided additional experimental data to support this conclusion in Figure 6 S2 showing ITC with an H2 specific construct of IC (IC_37-88_) that binds weakly with p150_CC1B_ but does not bind NudE_CC_.

2) ITC, NMR, and AUC data are presented for the binding of NudE to IC(1-260). Some more clarification is needed in terms of the interpretation of these data, also in the context of the observations on IC(FL). The experimental observations do not seem to be explainable simply by a weak complex or a concentration-dependent effect as suggested by the authors.

By AUC and ITC, NudE_CC_ exhibits binding to IC_1-88_ (Figure 6A, C). By ITC the dissociation constant of this interaction is moderate at 0.3 µM and by AUC, we see a less dramatic shift in sedimentation constant than expected, suggesting that either the interaction is dynamic or that binding induces a more extended structure for the complex. It is important to note that the sedimentation coefficient of a complex is not only influenced by size but also by shape and by binding on/off rates. NMR further clarifies that, unlike p150_CC1B_, NudE_CC_ does not interact with the H2 region of IC (Figure 6B). This important distinction plays a larger role in the interaction of NudE_CC_ with IC_1-260_ or IC_FL_ as we see that with the larger constructs binding is inhibited due to the tertiary, autoinhibitory contacts within IC.

By ITC, binding is not detected for NudE_CC_ and IC_1-260_ at the concentration used. By AUC, a peak for a complex with a larger sedimentation coefficient is not seen for NudE_CC_ and IC_1-260_, however the absence of a peak corresponding to free IC indicates that some degree of highly dynamic binding takes place. We believe this is due to a minor population of “open”, uninhibited IC_1-260_ that exists in solution. NMR captures these minor states of IC that we propose to be concentration independent (evidence of tertiary contacts present for apo IC_1-260_ in Figure 4). While the teritary contacts are not concentration dependent, the population of these minor states are simply higher at higher concentration giving rise to what we see as some binding.

3) The difference in sedimentation coefficient of the dynein subcomplex + NudE and of the dynein subcomplex + p150 is surprisingly large suggesting significantly different shapes of the two bound complexes. Some discussion of this issue is present in the manuscript, but no clear explanation is provided. It would seem necessary to confirm these observations by other complementary techniques. With respect to the SV-AUC data, it would be helpful to use frictional coefficients f/F0 to disentangle Mw and conformation from sedimentation coefficients.

We were also surprised by the large difference in sedimentation coefficient between the dynein subcomplex with NudE_CC_ compared to the subcomplex with p150_CC1B_. We propose a difference in the level of elongation/compaction between the two subcomplexes as an explanation. As suggested by the reviewer, we have added calculated f/F0 values for all three variations of subcomplex (based on assumed complex molecular weights) and have added them into the results and Discussion sections. We have also better explained and incorporated earlier results from our research group (Morgan et al. 2021) that employ NMR PRE experiments and shows that N-IC exists as an equilibrium mixture of extended and compact states and that binding of NudE shifts the equilibrium toward more open states (which results in a larger f/F0 value and smaller sedimentation coefficient). We have also included SEC-MALS data (Figure 6 S1) of IC_1-88_ bound to p150_CC1B_ that eliminates the possibility that more than a p150 dimer binds to IC, reaffirming that it is very unlikely that the surprisingly large AUC sedimentation coefficient of the dynein subcomplex with p150_CC1B_ is due to a larger mass/binding stoichiometry than expected.

Multiple sources of complementary data to support a shape difference between subcomplexes are already included in the manuscript. For example, by SEC the dynein subcomplex with p150_CC1B_ elutes earlier – also indicating a difference in shape/migration compared to the dynein subcomplex with NudE_CC_. AUC results are also consistent throughout the manuscript, regardless of length of the IC construct used (Figure 6 & 9). We have attempted negative stain on the subcomplexes to further deconvolute the differences in shape/confirmation but have thus far been unsuccessful as the complexes dissociate at the concentrations needed for EM grids.

4) The authors suggest that the binding of LC7 releases the auto-inhibitory interaction of IC, however, NMR does not directly support this conclusion (Figure 7B). Some discussion of why this long-range interaction inhibits the binding of NudE, but not LC7 itself, should be included.

Edits have been made to the discussion to better explain why LC7 binding is not inhibited in the same manner as NudE_CC_. Briefly, LC7 is a globular protein and its binding site for IC is more complex than the binding site for IC on NudE. The LC7 binding site on IC is longer than the corresponding SAH region it covers in our autoinhibition model (Figure 10 – updates have been made to show the LC7 site as more exposed). In Figure 7B, IC_1-260_ binds to both LC7 and NudE_CC_ due to high protein concentrations and the unmatched sensitivity of NMR to binding interactions that likely captures the “open” minor states of IC.

5) As indicated in the summary above more care should be placed on introducing the system. For example, it was felt that Figure 1B, did not help much (e.g. the light chains were hard to acknowledge as they appear to be rather small compared to the IC chain and were at first overlooked at just the binding sites; where is the heavy chain of dynein, why is there no coiled coil of p150, etc?). In a related manner, the functional role of autoinhibition should be more clearly emphasized in the discussion, e.g., what is the relevance of the differential binding of the two ligands and their differential effects on the autoinhibited state. Which biological outcomes are to be expected?

Edits have been made to Figure 1 with special care to panel B. For example, the light chains and IC have been drawn more to scale, each protein subunit of interest is now clearly labeled, the helices of IC are included, and the coiled-coil of p150 has been changed to match the coiled-coil representation used for NudE. Subunits of dynein that are of less interest for this work are purposefully shown more crudely, and this is now better acknowledged in the figure’s caption.

A more thorough introduction of dynein is now included in the introduction, including a highlight on diseases caused by dynein mutations and edits have been made to introduce the IC subcomplex more clearly. A clearer connection between disorder and regulation is also now included.

The discussion has been edited for length, eliminating repetition of what is included in Results sections and, most importantly, now ends with a much stronger overview of dynein autoinhibition and the biological relevance of the findings of our work.

6) One of the conclusions is that the internal contacts occur between the C-terminal of the IC and SAH/H2, which is seen from the intensity changes in the HSQC spectra upon addition of the 160-240 construct to the 1-88 construct. However, adding the linker part from 100-160 produces a much more pronounced effect (Figure 5C, bottom), suggesting that residues in this region, which includes the Tctex and LC8 binding motifs play additional roles. Is the binding of the light chains to IC of higher affinity to the 100-160 protein than to the 1-260? In that case, this could suggest that also inhibitory access to these two sites occurs in the autoinhibited state. The additional effect of the 100-160 residues should be addressed.

We appreciate the question of whether the LC8 and Tctex binding sites are also inhibited, however we want to note that if this is the case, the effect is not pronounced enough to be seen by AUC experiments conducted with IC_1-260_ or IC_FL_. Because it does not appear that LC8 and Tctex are inhibited in any of the AUC experiments we do not find it necessary to study the binding affinities further with ITC. Furthermore, residues 100-150 do not appear to impact p150_CC1B_ and NudE_CC_ binding (Figure 6C, middle panels) and so the additional effects of 100-160 although noteworthy, are not investigated further.

7) Can H3 be excluded as a player in the internal interactions, just because you see binding to the LC7 site when studied in isolation? Once the LC7 regions is bound, H3 may also participate, as also clearly indicated from the data shown in Figure 4A. Using an H3 peptide would be relevant.

We agree that H3 cannot be completely excluded as a player in the internal interactions of IC, however, because the IC_160-240_ construct which contains H3 showed minimal evidence of binding (Figure 5) we would not expect an H3 peptide to show different results. Figure 4A shows that there are no assignable peaks for the H3 region, but this does not automatically mean that H3 is involved in the internal interactions. It is common for helical regions to also have lower NMR peak intensities compared to disordered regions, which is another potential explanation for why H3 peaks are either missing or unassignable. Further, we did not focus further on H3 because it is specific to CT-IC and not observed in ICs from other species.

8) Fig6B and associated text: there is a clear although weak loss in intensity/peak volume in the H2 region for the interaction with NudE. Why assume that there is no interaction? The affinity for NudE is lower, so the concentration of the complex will also be lower at similar conditions compared to that of p150, and this would give rise to the smaller effects in the spectra. In the lower panel, there is a clear indication of binding to H2 as well, and SAH and H2 binding may very well be cooperative as they are sequentially close. What are the relative concentrations of NudE and p150 in the cell? Would they be competitive despite the difference in affinities? Can a mechanism for p150 ability to relieve autoinhibition be proposed – from Fig6B, could it be able to bind first to the H2 region even though SAH is involved in the autoinhibitory interaction?

We have added further experimental evidence in Figure 6 S2 from ITC showing that binding is not detected between NudE_CC_ and H2 whereas weak binding is seen for p150_CC1B_ and H2. As now is more clearly stated in the text, the disappearance of IC_1-88_ peaks for both the SAH and H2 regions with NudE_CC_ can be explained by an interaction between the SAH and H2 regions identified by previously performed NMR paramagnetic relaxation enhancement (PRE) experiments in *Drosophila* IC. This inherent SAH/H2 interaction may cause some level of conformational exchange broadening when both the SAH and H2 regions are present, although it is clear from our data that H2 is not the driver of NudE/IC binding.

We do propose, as illustrated in Figure 10, that the weak binding between p150_CC1B_ and H2 allows for p150 _CC1B_ to bind IC despite the autoinhibition of the SAH region. As this binding is weak (~21 µM) we propose that IC autoinhibition is not completely relieved.

We were unable to find documentation comparing the relative concentrations of NudE and p150 in the cell and are therefore unable to comment on their competitiveness related to their affinities. This subject is further complicated by the fact that p150_CC1B_ is a subunit of dynactin, a protein complex that binds to dynein independently of the interaction between p150_CC1B_ and IC via other coiled-coil “activator” proteins.

[Editors' note: further revisions were suggested prior to acceptance, as described below.]

The manuscript has been improved but there are some remaining issues that need to be addressed, as outlined below:Reviewer #1 (Recommendations for the authors):I have received the revised version of the manuscript by Jara et al. Overall, the authors have done a good job in the revision process, in particular by adding new data to support their conclusions. I am overall in favour of publishing the manuscript in eLife.One point where I still remain less convinced is about the origin of the lack of direct correlation between the ITC, NMR and AUC data. The authors suggest that these techniques are differentially sensitive to protein concentration and to the presence of minor/open and auto-inhibited states of IC. This could be true; however, only a little evidence for this is actually provided.

We appreciate the reviewer’s concern regarding the origin of the lack of direct correlation between ITC and NMR, but this discrepancy can be easily explained by the different concentrations required for each technique. You see evidence of binding at concentrations above the Kd (such as with the NMR), and for the ITC experiments, the concentrations possible are below the Kd. It is difficult to redo the ITC at NMR concentrations because these are protein-protein interactions with the protein in the syringe requiring to be about 10 times more concentrated than the protein in the cell, while for the NMR experiments, both proteins are mixed at similar concentrations and then the mixture is concentrated.

I hope this is explanation is now sufficient.

Reviewer #3 (Recommendations for the authors):The authors have thoroughly revised the manuscript and included further evidence to support their conclusion, and the relevance of the data for the biology of the protein complex is now addressed to some extent in the discussion.A few comments remain:Figure 1 has improved but can be further improved by adding N – and C to the IC ends, indicate the HC (heavy chain), LIC (dynein light intermediate chain), N-IC (N-terminal domain of IC), C-IC, SAH (single α helix), H2 (short helix), all of which are mentioned in the text. Perhaps use a zoom on the region central to the study. Also, the use of CT-IC is confusing, and the authors should use more correct CtIC to indicate the reference to the species.

The recommended changes to Figure 1 have been considered carefully. Ultimately, the addition of more labels to panel B take away from the readability. The LIC is purposefully left out of any special recognition in the schematic, and the figure caption has been made clearer. We believe panel A acts as a key for panel B as Ct IC is shown in the same orientation with the N-terminus to the left. The use of “CT-IC” has been replaced by “Ct IC” throughout.

Although I agree with the authors that the interaction with NudE is dominated by the SAH interaction, some interaction with H2 is seen in the data in Figure 6b, with some intensity changes for isolated H2 binding, and loss of signals for H2 in the 1-88 construct, and with the linker residues between SAH and H2 still detectable. This adds to the explanation of the larger binding affinity in the longer construct. The weak engagement of H2 with NudE may allow p150 to invade a NudE:IC complex, possibly in the trimeric complex (see also work by Berlow et al. (Nature 2017 and follow up paper)) and may be of biological relevance for regulation. This offers an alternative interpretation of the data, p. 12, l. 268-271, also relevant for discussion p. 17, l. 405-406, and p. 417.

Wording of a possible weak engagement between NudE and H2 has been added an alternative explanation to the NMR data presented in Figure 6b. As far as the reviewer’s suggestion of a trimeric complex in the competition between NudE and p150, we have reported such competition in prior work (Nyarko et al., 2012) that is now included and cited in our discussion.

The auto-inhibited state may be further stabilized. Data in Figure 5 suggest interaction between SAH and the LC7 (as noted by the authors), but also additional interaction between H2 and H3 (4C-top compared to 4C bottom), not mentioned.

We appreciate this comment and eye to detail and have now mentioned this additional interaction in the discussion text. After careful consideration, we chose not to depict interaction between H2 and H3 in our final model (Figure 10) as to not further complicate an already complex model. In addition, because H3 is specific to Ct and not present in other ICs, we considered it not essential to highlight in our model.

I appreciate the inclusion of the ITC data, and the two-state binding fit, but the data is not discussed. With N-values of 0.5 and a dominantly entropically driven interaction, how is this interpreted by the authors.

Further clarification has been added to the text to include the note that all p150CC1B isotherms were fit with a two-step binding model to provide the best possible fits, resulting in N values of 0.5, purposely imposed for longer IC constructs based on known overall 1:1 binding stoichiometry (Figure 6-figure supplement 1). In the discussion, we have now also included interpretation of this entropically driven interaction as opening of IC, necessary for binding.